# Mode-phase-difference photothermal spectroscopy for gas detection with an anti-resonant hollow-core optical fiber

Pengcheng Zhao [1,2,3], Yan Zhao [2,3], Haihong Bao [2,3], Hoi Lut Ho [2,3], Wei Jin[2,3 ✉], Shangchun Fan[1,4 ✉], Shoufei Gao [2,5], Yingying Wang [5] & Pu Wang[5]

Laser spectroscopy outperforms electrochemical and semiconductor gas sensors in selectivity and environmental survivability. However, the performance of the state-of-the-art laser sensors is still insufficient for many high precision applications. Here, we report mode-phase-difference photothermal spectroscopy with a dual-mode anti-resonant hollow-core optical fiber and demonstrate all-fiber gas (acetylene) detection down to ppt (parts-per-trillion) and <1% instability over a period of 3 hours. An anti-resonant hollow-core fiber could be designed to transmit light signals over a broad wavelength range from visible to infrared, covering molecular absorption lines of many important gases. This would enable multi-component gas detection with a single sensing element and pave the way for ultra-precision gas sensing for medical, environmental and industrial applications.

[1] School of Instrumentation and Optoelectronic Engineering, Beihang University, Beijing 100191, China. [2] Department of Electrical Engineering and Photonics Research Center, The Hong Kong Polytechnic University, Hung Hom, Kowloon, Hong Kong, China. [3] Photonics Research Center, The Hong Kong Polytechnic University Shenzhen Research Institute, No. 18 Yuexing 1st Road, Nanshan District, Shenzhen 518057, China. [4] Key Laboratory of Quantum Sensing Technology (Beihang University), Ministry of Industry and Information Technology, Beijing 100191, China. [5] Beijing Engineering Research Centre of Laser Technology, Institute of Laser Engineering, Beijing University of Technology, Beijing 100124, China. ✉email: eewjin@polyu.edu.hk; shangcfan@buaa.edu.cn

Ultrasensitive detection of molecular gases has important applications in atmospheric and planetary science[1,2], breath diagnostics[3], and combustion monitoring[4]. Comparing with semiconductor and electrochemical sensors, laser absorption spectroscopy (LAS) offers better selectivity[5]. The sensitivity of LAS depends on absorption path length and line strength that is weak in the near infrared (NIR) and much stronger in the mid-infrared (MIR). With tunable diode laser absorption spectroscopy (TDLAS) and multi-pass Herriott cells to achieve long effective path length, sub-ppm (parts-per-million) and ppt (parts-per-trillion) level noise-equivalent concentration (NEC) have been achieved, respectively, in the NIR and MIR[6,7]. One of the most sensitive spectroscopic techniques is the noise-immune cavity-enhanced optical heterodyne molecular spectroscopy (NICE-OHMS), which could achieve ppq (parts-per-quadrillion) level NEC in the NIR[8,9]. However, there is a trade-off between sensitivity and dynamic range. With a cavity finesse of 30,000, the dynamic range is limited to about four orders of magnitude[10]. In addition, for cavity-enhanced systems with long effective path lengths (e.g., several kilometers for NICE-OHMS), complex cavity-locking using, for example, the Pound-Drever-Hall technique is needed and it would be difficult to maintain the performance over a long period of time in a real-world environment.

Photoacoustic spectroscopy (PAS) and photothermal spectroscopy (PTS) measure, respectively, the pressure and refractive index (RI) change owing to optical absorption. PAS does not require a long optical path length and has demonstrated ppt-level NEC for acetylene by combining an optical power build-up cavity with cantilever-enhanced detection in the NIR[11]. With hollow-core optical fiber (HCF) gas cells of moderate optical path lengths (i.e., meters), PTS has demonstrated ppb-level NEC in the NIR[12,13]. Operating in the NIR would allow the use of mature fiber-optic components, which have the advantage of compactness, remote interrogation, and multi-point detection[14–17].

Here, we report mode-phase-difference (MPD) PTS for high performance gas detection. The MPD between two transverse modes of a gas-filled HCF is measured interferometrically and taken as the photothermal (PT) signal. The MPD is sensitive to gas absorption in the hollow-core but insensitive against external environmental perturbations, enabling remarkably high signal-to-noise ratio (SNR). With meters-long HCF gas cells operating in the NIR telecom band, we demonstrate all-fiber optical acetylene detection with ppt level NEC and an unprecedented dynamic range of seven orders of magnitude.

## Results

**Theory**. MPD-PTS uses a dual-mode HCF as the gas cell. Figure 1a shows a single-ring (SR) anti-resonant (AR) HCF used in this research. It has a broad transmission band from below 850 to beyond 1700 nm and supports a fundamental $LP_{01}$-like mode and a second-order $LP_{11}$-like mode, as shown, respectively, in Fig. 1b, c.

The basics of MPD-PTS may be explained intuitively by using the illustrations in Fig. 2. A modulated pump laser beam propagating in a dual-mode HCF is absorbed by trace molecules in the hollow-core, which heats up the gas and perturbs the RI distribution. The heating profile follows the intensity distribution of the pump, which depends on the fractional power of the pump in the two modes. The pump intensity would have an approximate Gaussian distribution for the $LP_{01}$ mode and a two-lode distribution for the $LP_{11}$ mode. Ideally, all the pump power should be launched into the $LP_{01}$ mode (Supplementary Note 2) and the heating profile in the cross-section would be approximately Gaussian and invariant along the HCF. Figure 2a shows a general scenario where both $LP_{01}$ and $LP_{11}$ modes are

present in the HCF. The pump intensity, the temperature (and hence the RI) distribution in the hollow-core varies periodically along the HCF owing to coherent mixing of the two modes and the spatial period equals to the modal beat length $l_b$ (Supplementary Note 2). Figure 2b, c show, respectively, the variations of pump intensity and temperature along the HCF. If a probe laser beam is now propagating simultaneously in the HCF, it will be modulated by the RI change. The probe also has $LP_{01}$ and $LP_{11}$ modes, and the PT phase modulations for the two probe modes are different, depending on the overlap integral of the mode fields with the RI perturbation, as shown in Fig. 2c. According to the Beer–Lambert law and under the assumption of weak absorption, i.e., $\alpha(\lambda_{pump})CL << 1$, the differential PT phase modulation is proportional to gas concentration $C$ and given by

$$\delta\phi = \Delta\phi_{01} - \Delta\phi_{11} = k^*(\eta, f)\left(1 - e^{-\alpha(\lambda_{pump})CL}\right)P_{pump}$$
$$\approx k^*(\eta, f)\alpha(\lambda_{pump})CLP_{pump} \qquad (1)$$

where $\Delta\phi_i$ ($i = 01$ or $11$) represents phase modulation for $LP_{01}$ or $LP_{11}$ mode of the probe, $\alpha(\lambda_{pump})$ the absorption coefficient for a relative concentration of 100%, $\lambda_{pump}$ the wavelength of the pump, $L$ the length of the sensing HCF, $P_{pump}$ the average pump power over $L$, $\eta$ the fractional pump power in the $LP_{01}$ mode and $f$ the pump modulation frequency. The differential phase modulation coefficient $k^*$ is a function of $\eta$ and $f$. The detailed formulations of differential modulation and calculation of $k^*$ are presented in Supplementary Note 2. For $\eta = 90\%$ and $f < 10$ kHz, $k^*$ is determined to be 1.7 ($\pm 0.2$) $\times 10^{-7}$ rad cm ppm$^{-1}$ m$^{-1}$ mW$^{-1}$ for balance gas of argon at 1.5 bar. It is ~ 20% of the phase modulation coefficient of the fundamental mode for the SR-AR-HCF, which means that the MPD is still reasonably sensitive to gas absorption in the hollow-core. This is because that the fields of the probe $LP_{01}$ and $LP_{11}$ modes overlap differently with the non-uniform heating profile and the phase sensitivities of the two modes to gas absorption are quite different. The variation in the MPD can be conveniently detected with a proper in-line dual-mode fiber interferometer, as shown in Fig. 2d.

However, sensitivity of the MPD to external perturbation can be much smaller than that of the fundamental mode phase. Owing to the very small diameter of the HCF, the environmental (e.g., temperature and pressure) changes would result in uniform RI change across the hollow-core region and hence affect the phases of the $LP_{01}$ and $LP_{11}$ modes similarly, meaning that the MPD is insensitive to environmental perturbation. The sensitivities of the $LP_{01}$ mode phase and the MPD to external perturbation ($X$) may be expressed as[18,19]

$$\frac{\Delta\phi_{01}}{\Delta X} = \frac{2\pi n_{01}L}{\lambda}\left(\frac{1}{n_{01}}\frac{\partial n_{01}}{\partial X} + \frac{1}{L}\frac{\partial L}{\partial X}\right) \qquad (2)$$

$$\frac{\delta\phi}{\Delta X} = \frac{2\pi\Delta nL}{\lambda}\left(\frac{1}{\Delta n}\frac{\partial \Delta n}{\partial X} + \frac{1}{L}\frac{\partial L}{\partial X}\right) \qquad (3)$$

where $\Delta n = n_{01} - n_{11}$ with $n_{01}$ and $n_{11}$ representing, respectively, the effective RI of probe $LP_{01}$ and $LP_{11}$ modes. The sensitivity ratio $\zeta = \Delta\phi_{01}/\delta\phi$ is on the order of $n_{01}/\Delta n$, indicating that the MPD is much less sensitive to external perturbation than the fundamental mode phase. Numerical simulation with COMSOL Multiphysics (Supplementary Note 3) shows that the values of $\zeta$ are, respectively, ~280 for temperature and ~170 for pressure for the SR-AR-HCF at 1550 nm. To summarize, the MPD is reasonably sensitive to gas absorption in the hollow-core but very insensitive to external disturbances, which enhances the SNR significantly and makes extremely sensitive gas sensors possible.

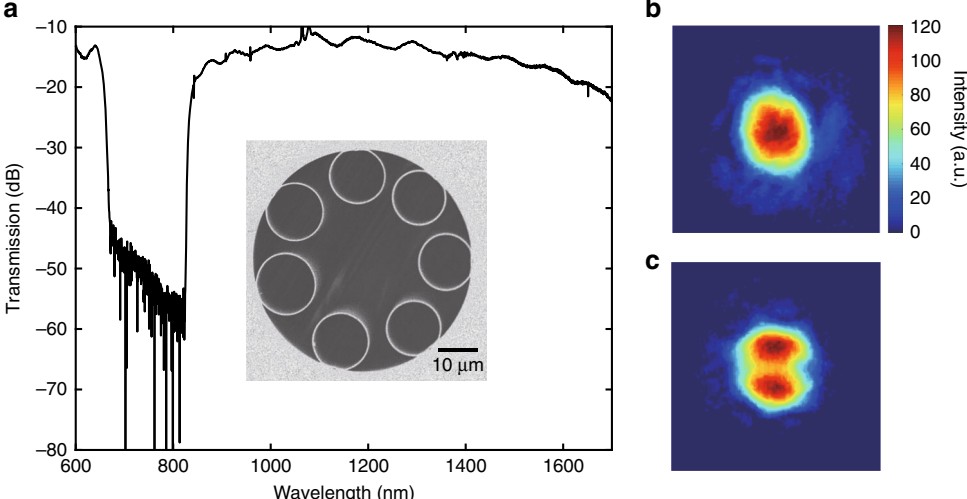

**Fig. 1 The HCF. a** Cross-section and spectral transmission of a 4-m-long SR-AR-HCF. The HCF has a silica outer cladding with inner diameter of ~56 µm, seven capillary rings with diameter of ~14 µm and thickness of ~370 nm, giving an inscribed air core with diameter of ~28 µm. **b**, **c** Intensity profiles of **b** the fundamental mode $LP_{01}$ and **c** a mix of the $LP_{01}$ and second-order $LP_{11}$ modes at 1550 nm, viewed at the output of a 1.5-m-long SR-AR-HCF. The orientations of the mode patterns are stable and do not vary with small perturbation applied to the HCF. By varying launch condition, another $LP_{11}$ mode (not shown) with lobe-orientation orthogonal to the one shown in this figure can be excited, we called the two $LP_{11}$ modes $LP_{11a}$ and $LP_{11b}$, respectively. No other higher-order modes are observed when input launching condition is varied. The calculated field distributions of the $LP_{01}$ and $LP_{11}$ modes are presented in Supplementary Note 1. The effective RI difference between the $LP_{01}$ and $LP_{11}$ modes is ~$1.2 \times 10^{-3}$.

**Experimental setup**. Two gas cells are made using the SR-AR-HCF shown in Fig. 1a. One uses a 4.67-m-long HCF with both ends mechanically spliced to single mode fiber (SMF) pigtails. A small air-gap is kept at the spliced joints for gas filling into the HCF. The other uses a 0.74-m-long HCF with both ends mechanically spliced to SMF pigtails, and 34 lateral micro-channels are fabricated along the HCF for fast gas filling. Lateral offsets are introduced at the splicing joints to excite and to collect light from the $LP_{01}$ and $LP_{11}$ modes to form an in-fiber dual-mode interferometer to probe the modulation in the MPD. The structures, fabrication processes, and properties of the HCF gas cells can be found in Methods.

The experimental system is shown in Fig. 3. The pump source is a 1.53-µm distributed feedback laser and its wavelength is modulated sinusoidally at ~9.5 kHz via the internal signal generator of lock-in amplifier (LIA) and at the same time scanned slowly across the P(13) absorption line of acetylene at ~1532.830 nm. The amplitude of wavelength modulation is set to be ~2.2 times the linewidth of the P(13) line to maximize the second harmonic ($2f$) signal[20]. The probe source is an external-cavity diode laser and its wavelength is tuned to a quadrature point of the dual-mode interference fringe at ~1550 nm. The pump and probe beams are launched into the HCF from the opposite directions via two wavelength-division multiplexers (WDM1 and WDM2), and the probe beam coming out from WDM2 is detected by a photodetector (PD). The $2f$ component of the MPD modulation is demodulated by the LIA and transferred to a computer via a data acquisition (DAQ) card for further processing.

**Lower detection limit and long-term stability**. Experiments were first conducted with the 4.67-m-long HCF gas cell filled with the calibrated gas sample of 1 ppm acetylene in argon. The gas was pressurized into the hollow-core with ~1.95 bar pressure at the input joint, whereas the output joint is open to atmosphere. The details of the experiments can be found in Supplementary Note 5. Figure 4a shows the $2f$ lock-in output (the PT signal or $2f$ signal) when the pump is tuned across the P(13) line of acetylene

for different pump power levels delivered into the HCF. The $2f$ lock-in output when the pump is tuned away from the absorption line to ~1532.57 nm is also recorded and regarded as background noise. The PT signal increases linearly with pump power, whereas the standard deviation (s.d.) of the noise remains almost unchanged, as shown in Fig. 4b. At the pump power of ~108 mW, the average SNR over 10 measurements is ~14809 for a lock-in time constant of 1 s with 18 dB/Oct slope, giving a NEC of ~68 ppt for SNR of unity (i.e., $1\sigma$).

Allan–Werle deviation analysis[21,22] is conducted with the noise data over a period of 2 hours, and the results are shown in Fig. 4c. The $1\sigma$ NEC obtained from the Allan–Werle plot is ~43 ppt and ~15 ppt, respectively, for 10 and 100 s averaging time. The system is capable of averaging over a much longer time of up to a few thousand seconds owing to superb noise cancellation ability of the MPD detection technique, which would enable sub-ppt level acetylene detection. This result is more than two orders of magnitude better than the previous optical fiber gas sensors[12].

The system stability, which is important for practical applications, is tested by repeatedly scanning the pump wavelength across the P(13) line and the results for 1 ppm $C_2H_2$ with a pump power of 108 mW are shown in Fig. 5. The $p$–$p$ value of DC compensated $2f$ signal varies ~0.8% over a period of 3-hours (see Supplementary Note 5).

**Dynamic range and response time**. With the same experimental setup but the 0.74-m-long HCF gas cell, the dynamic range of the sensing system is tested by filling different concentrations of acetylene into the hollow-core. Figure 6a shows the $2f$ signal for 4, 10, 20 ppm acetylene in nitrogen, whereas Fig. 6b shows the $p$–$p$ value of the $2f$ signal as functions of acetylene concentration from 4 ppm to 4% acetylene in nitrogen at atmospheric pressure. An approximately linear relationship is obtained for acetylene concentration up to ~1% and significant non-linearity starts to appear beyond this value. The $1\sigma$ NEC is evaluated by filling the calibrated 1 ppm acetylene in argon at the pressure of ~1.5 bar into the HCF and found to be 2 ppb for 1 s lock-in time constant. Alan–Werle deviation analysis (see Supplementary Note 5)

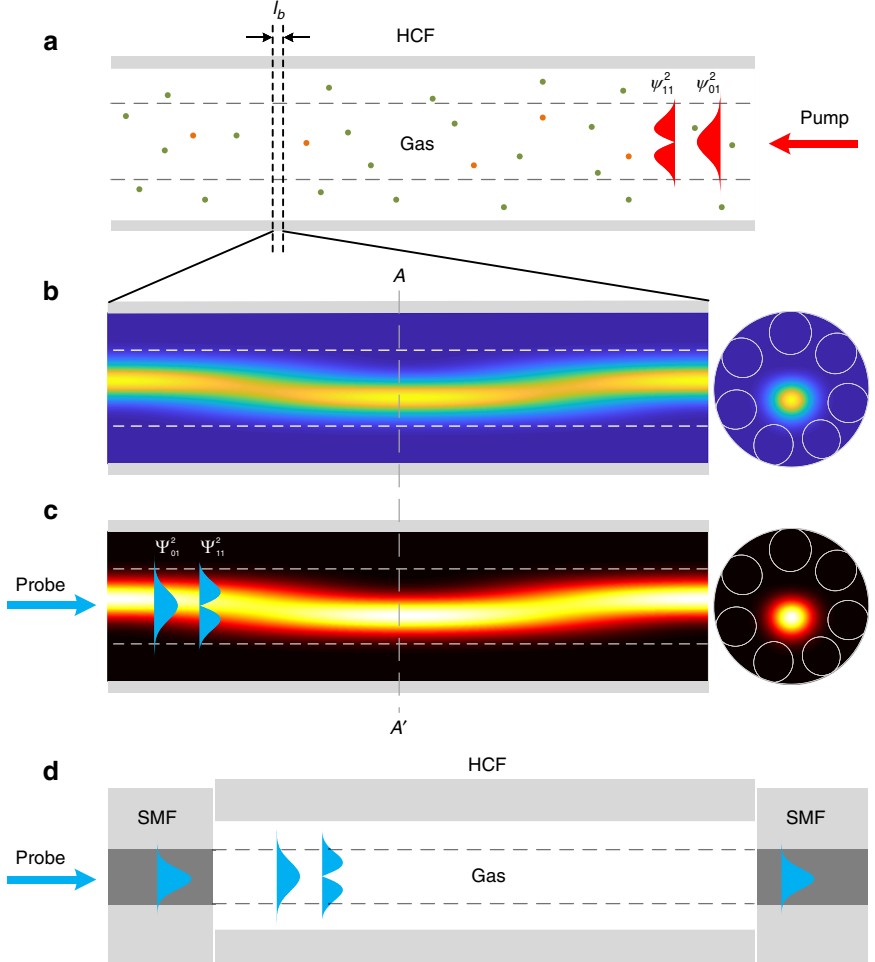

**Fig. 2 Principle of MPD-PTS. a** A schema showing gas in the hollow-core and pump intensity profiles of individual LP$_{01}$($\psi_{01}^2$) and LP$_{11}$($\psi_{11}^2$) modes. **b** Variation of pump intensity over a modal beat length $l_b \approx 1.3$ mm for $\eta = 90\%$. The panel on the right shows the intensity at the cross-section A–A'. **c** The temperature (hence RI) profile over the same length of HCF. The panel on the right shows the temperature distribution over the cross-section A–A'. The probe intensity profiles of LP$_{01}$($\Psi_{01}^2$) and LP$_{11}$($\Psi_{11}^2$) modes are superimposed onto the temperature profile to show the different overlap between them. **d** A SMF-HCF-SMF dual-mode interferometer for detecting variation in the MPD.

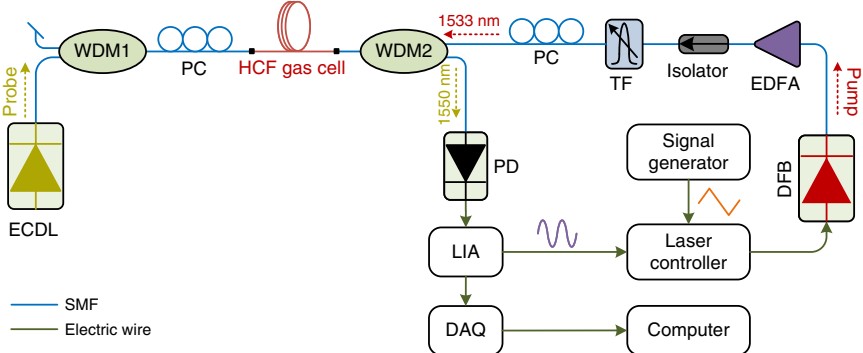

**Fig. 3 Experimental setup.** *EDFA* erbium-doped fiber amplifier, *TF* tunable optical filter to minimize EDFA's amplified spontaneous emission noise, WDM1 and WDM2 1550 nm/1530 nm wavelength-division multiplexers, *PC* polarization controller, *LIA* lock-in amplifier, *DAQ* data acquisition card, *DFB* distributed feedback laser (the pump), *ECDL* external-cavity diode laser (the probe), *PD* photodetector.

obtains NEC of ~500 ppt for an averaging time of ~100 s, giving a dynamic range of ~$2 \times 10^7$, nearly two orders of magnitude larger than the state-of-the-art gas detection systems[12].

The response time is tested by filling HCF gas cell, via the microchannels, with nitrogen, 1000 ppm acetylene in nitrogen and then nitrogen. For this experiment, the pump laser wavelength is fixed to the center instead of scanning across the P(13) line. The *2f* signal from the LIA is shown in Fig. 7. The response time $t_{90}$, which is defined as the time taken to reach 90% of the applied concentration[23], is ~44 s, showing that the introduction of microchannels breaks limitation of slow response in HCF gas sensors[24].

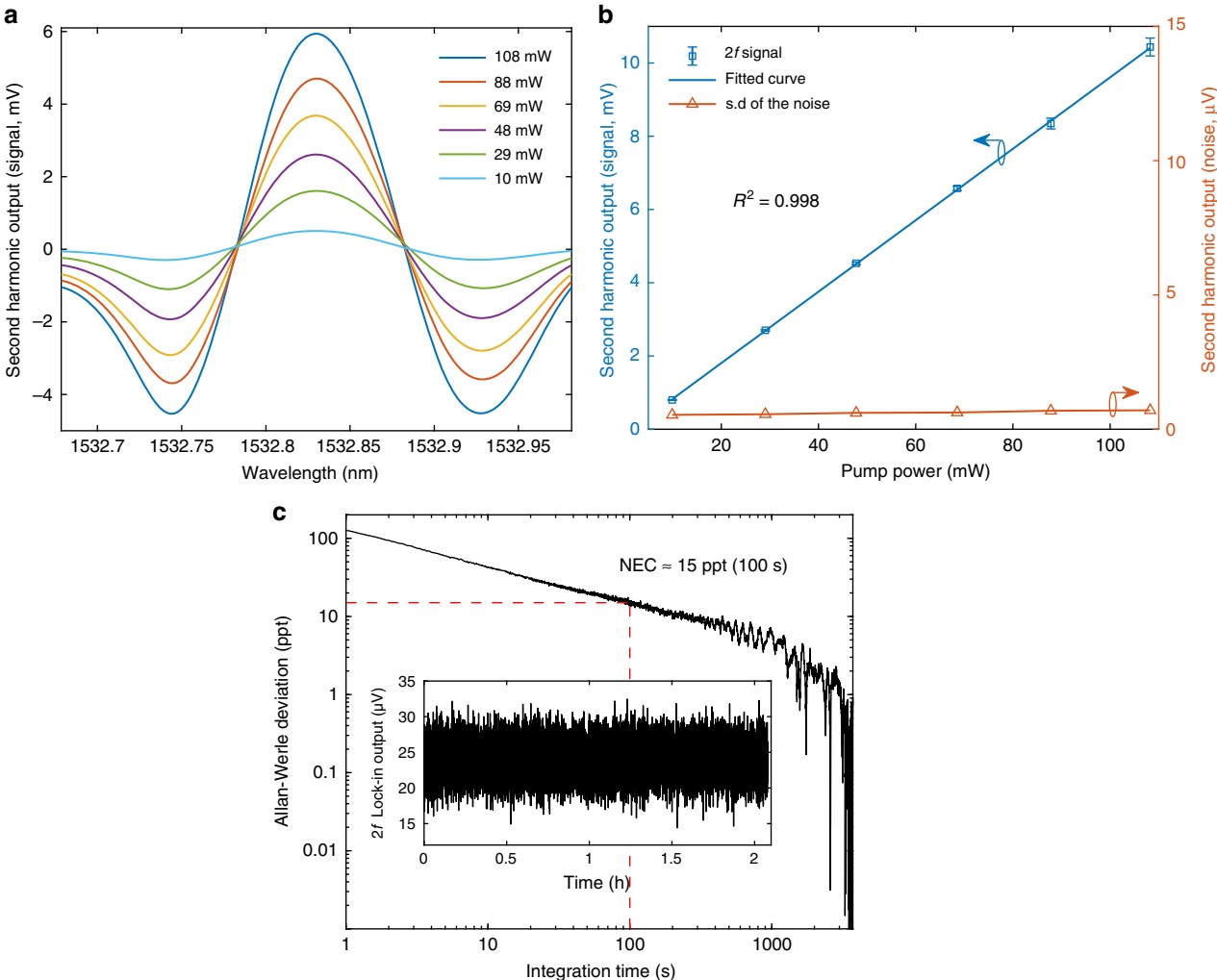

**Fig. 4 Results of gas detection with the 4.67-m-long HCF gas cell. a** The PT signal (2$f$ signal) with pump wavelength tuned across the P(13) line of $C_2H_2$.
**b** The peak-to-peak ($p$–$p$) value of the 2$f$ signal and the s.d. of the noise as functions of pump power level. Error bars show the s.d. from 10 measurements and the magnitudes of the error bars are scaled up by fivefold for clarity reason. Source data are provided as a Source Data file. The mean probe power on the PD is −9 dBm. The lock-in time constant is 1 s and filter slope is 18 dB Oct$^{-1}$, corresponding to 0.094 Hz detection bandwidth. **c** Allan–Werle plot based on the noise data over a period of 2 hours, which is shown in the inset. For the noise measurement, the lock-in time constant is 100 ms, corresponding to 0.94 Hz detection bandwidth.

## Discussion

The PT signal (i.e., 2$f$ signal) increases linearly with pump power, whereas the noise level stays almost unchanged. This leaves rooms for further improving the detection sensitivity by simply increasing the pump power level. The overlap of mode fields with fiber material (silica) in the SR-AR-HCF is extremely small, meaning that the HCF would have a high damage threshold. In fact, transmission of laser beams with average power as high as 70 W has been demonstrated with a SR-AR-HCF[25]. If we could deliver ~1 W pump power into the SR-AR-HCF by, for example, using a large-mode-area fiber without an air-gap at the fiber joint, it would be possible to achieve sub-ppt level acetylene detection. For 1 W pump power launched into SR-AR-HCF filled with 5% $C_2H_2$ at ambient conditions, the highest temperature rise is calculated to be ~14 °C, well below acetylene's autoignition temperature of 305 °C according to DIN EN 14522-2005. The pump power density for 1 W in the hollow-core is ~$10^5$ W/cm$^2$, far below the breakdown value ~$10^{12}$ W/cm$^2$ of air, which is similar to that of $C_2H_2$[26].

The long-term stability of the system benefits from two factors: (1) the SMFs are not parts of the interferometer for the MPD detection and hence disturbance on the SMFs would not affect the phase difference between the interfering modes, resulting in a better stability over the traditional two-beam interferometers such as Mach–Zehnder. (2) Comparing with the fundamental mode phase, the MPD is much insensitive to external disturbance. For the stability test results in Fig. 5, the probe wavelength is simply tuned to a quadrature point of the interference fringe. Stable operation over a much longer term may be achieved by locking the probe laser wavelength to a fringe quadrature via servo-control, and by using polarization maintaining SMFs for light transmission and a dual-mode sensing HCF capable of maintaining polarization as well as lobe-orientation[27].

The dynamic range of the system is limited by the non-linear transfer function of the interferometer as well as the non-linear dependence of the phase modulation beyond the weak absorption approximation[12]. With 110 mW pump power and the modulation frequency of 9.5 kHz, the differential phase modulation for the 0.74-m-long SR-AR-HCF filled with 1% acetylene in nitrogen is calculated to be ~0.14 rad, well within the 1% linear dynamic range of the interferometer transfer function (that is, $\delta\varphi \leq 0.1\pi$). Nevertheless, according to the Beer–Lambert law, the pump

power is absorbed significantly and reduced to ~50% at the output of the HCF gas cell, suggesting weak absorption assumption becomes invalid, which bears the primary responsibility for the observed non-linear behavior (See Supplementary Note 6).

The response time of the present system is not actually limited by the HCF gas cell, but by the time taken to fill the gas chamber. We placed the 0.74-m-long HCF gas cell inside a tubular chamber with the diameter of 25 mm and length of 1 m, and time is needed to fill gas into the chamber. With multiple microchannels fabri-

cated along the HCF, the response time limited by gas diffusion into the hollow-core can be as short as 3 s[24].

The performances of some state-of-the-art spectroscopic laser gas sensors are summarized in Table 1. For ease of comparison, we list the noise-equivalent absorption (NEA), which is independent gas types and absorption strength. The values of NEA can be converted from NEC through equation $NEA = NEC \cdot \alpha(\lambda_{pump})$. The NEA of the MPD-PTS is more than two orders of magnitude better than the state-of-the-art PTS system with a bandgap HCF[12] and comparable to some of the most sensitive spectroscopic techniques. However, the MPD-PTS system is much simpler and in an all-fiber format. The dynamic range is orders of magnitude larger than any of the previous techniques. The MPD-PTS system can operate under the condition of ambient pressure and temperature.

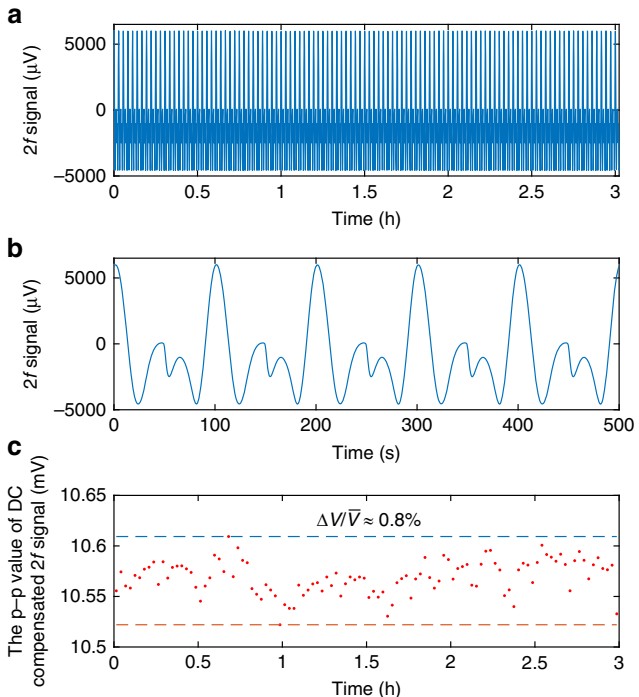

**Fig. 5 Results of long-term stability test. a**, **b** 2*f* signal **a** over a period of 3 hours and **b** from 0 to 500 s. **c** The variation of *p–p* value of DC compensated 2*f* signal over 3 hours.

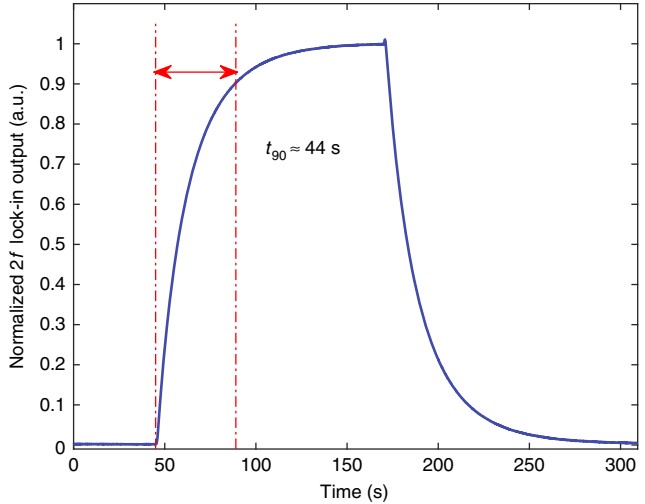

**Fig. 7 Results of response time test.** Normalized 2*f* signal recorded during gas loading. The nominal laser wavelength is tuned to the center of the P (13) line and at the same time modulated sinusoidally at 9.5 kHz. At ~45 s, 1000 ppm acetylene gas was loaded into the gas chamber with the differential pressure of ~1 bar. At ~170 s, nitrogen was loaded into the gas chamber.

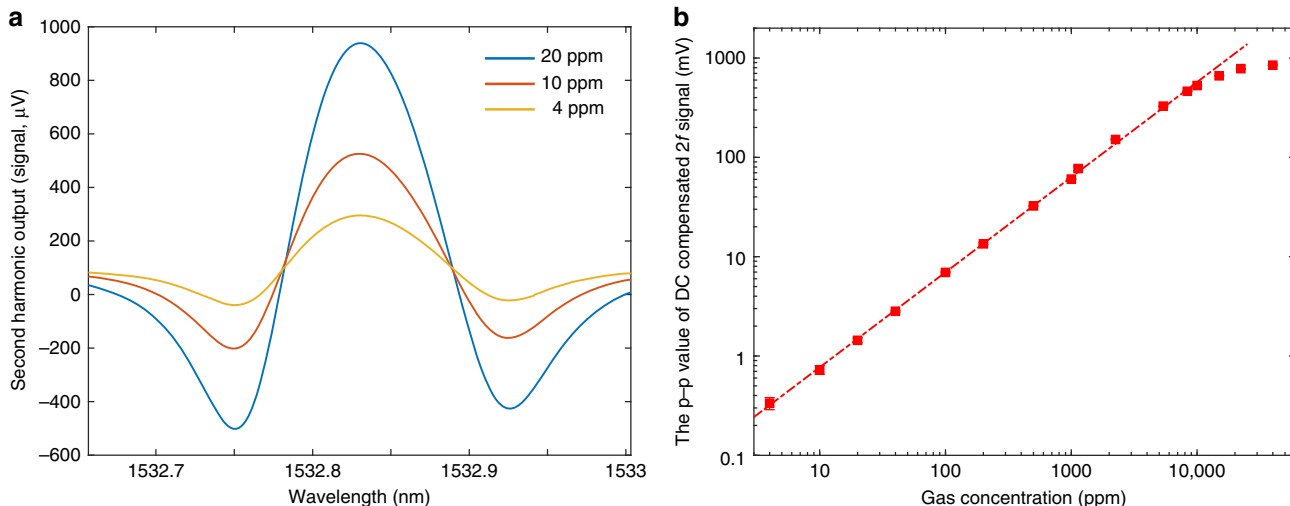

**Fig. 6 Results of dynamic range test. a** 2*f* -signal when pump laser is tuned across the P(13) line of acetylene for 4, 10, 20 ppm acetylene concentration. **b** The *p–p* value of the 2*f* signal as functions of gas concentration. Error bars show the s.d. from five measurements and the magnitudes of the error bars are scaled up by 50-fold for clarity reason. The mean probe power at PD is ~9.5 dBm. The time constant of the lock-in amplifier is 1 s for gas concentration below or equal 20 ppm and is 100 ms for above 20 ppm.

**Table 1 Performance of state-of-the-art spectroscopic laser gas sensors.**

| Gas type | Technique | Wavelength/μm | Effective path length/m | Dynamic range | NEA/cm$^{-1}$ |
|---|---|---|---|---|---|
| H$_2$S | TTFMS[31] | 1.577 | 1 | >68 | $5.3 \times 10^{-9}$ |
| NO | TDLAS[32] | 5.263 | 210 | Not stated | $1.5 \times 10^{-10}$ |
| CH$_4$ | OF-CEAS[33] | 2.33 | ~10,000 | Not stated | $1 \times 10^{-9}$ |
| CH$_4$ | CRDS[34] | 1.651 | ~12,000 | $> 5.2 \times 10^3$ | $2.3 \times 10^{-11}$ |
| C$_2$H$_2$ | NICE-OHMS[9] | 1.534 | ~13,796 | Not stated | $2.2 \times 10^{-14}$ |
| CO | OA-ICOS[35] | 1.565 | 2700 | Not stated | $1.9 \times 10^{-12}$ |
| C$_2$H$_2$ | CECEPAS[11] | 1.531 | ~ 0.15 | $> 6.7 \times 10^3$ | $2.8 \times 10^{-11}$ |
| C$_2$H$_2$ | PTS[12] | 1.53 | 10 | $5.3 \times 10^5$ | $2.3 \times 10^{-9}$ |
| C$_2$H$_2$ | MPD-PTS (this work) | 1.533 | 4.67 | $2 \times 10^7$ | $1.6 \times 10^{-11}$ |

*TTFMS* two-tone frequency modulation spectroscopy, *TDLAS* tunable diode laser absorption spectroscopy, *OF-CEAS* optical feedback cavity-enhanced spectroscopy, *CRDS* cavity ringdown spectroscopy, *NICE-OHMS* noise-immune cavity-enhanced optical heterodyne spectroscopy, the results were obtained with gas pressure of 50 mTorr, *OA-ICOS* off-axis integrated cavity output spectroscopy, *CECEPAS* cavity-enhanced cantilever-enhanced photoacoustic spectroscopy, *PTS* photothermal spectroscopy, *MPD-PTS* mode-phase-difference PTS, NEA is independent of absorption line strength and may be used to compare the performance of sensors for gases with different absorption line strength. The NEA data were calculated from NEC data and the absorption line strength from HITRAN database[36].

In summary, we developed a MPD PT spectroscopic technique for ultrasensitive gas detection over extremely large dynamic range. The detection of MPD minimizes the effect of environment perturbation while maintaining a sufficiently large PT modulation signal, enabling orders of magnitude of enhancement of SNR. With a 4.67-m-long HCF, we achieved detection limit down to low ppt level and 0.8% instability over a period of 3 hours. With a 0.74-m-long HCF, we demonstrated dynamic range of over 7 orders of magnitude and response time of ~ 44 s. The broadband transmission of the SR-AR-HCFs[28–30] will allow pumping at different wavelengths from visible to MIR, which, in combination with an ultra-low-noise fiber-optic probe interferometer in the NIR, would enable cost-effective, precision all-fiber sensors with multi-gas detection capability. The technique could also be applied to liquid-core fiber sensors and other dual-mode waveguide systems for the study of light-matter interaction.

## Methods

**Fabrication and characterization of HCF gas cells.** The 4.67-m-long HCF was made by mechanically splicing the SR-AR-HCF to an input SMF pigtail and an output SMF pigtail. For gas filling purpose, a gap of 1–2 μm was kept between the input SMF and the HCF, as well as between the HCF and the output SMF. A lateral offset of a few μm is introduced between the HCF and the probe input SMF, to excite the two probe modes (LP$_{01}$ and LP$_{11}$) simultaneously. The input mechanical joint is sealed within a T-shaped glass tube with a gas inlet and outlet. The HCF is also offset aligned to the output SMF and the relative lateral positions are adjusted by use of translation stages to achieve a reasonable fringe contrast for the interference between the two probe modes. For the 0.74-m-long HCF gas cell, the SR-AR-HCF is mechanically spliced to SMFs in the input and the output ends with lateral offsets to form an all-fiber modal interferometer. Microchannels are drilled, by use of a femtosecond infrared laser, from the side of the HCF, and gas filling into the hollow-core is achieved via these microchannels. For more details, please refer to Supplementary Note 4.

**Preparation of gas samples.** Gas samples with different concentrations are prepared by mixing calibrated gas samples at atmospheric pressure with two mass flow controllers. For gas concentration below 1000 ppm, the gas samples are prepared by mixing 1000 ppm acetylene in nitrogen with high-purity nitrogen with different ratios. For gas concentration above 1000 ppm, the samples are prepared by mixing high-purity nitrogen with 99.99% acetylene.

## Data availability

The data that support the findings of this study are available from the authors upon reasonable request.

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

## Acknowledgements

This work was supported by Hong Kong SAR government GRF grant (15220617), National Natural Science Foundation of China through NSFC grant (61827820, 61535004), The Hong Kong Polytechnic University (SB95, 1-ZVG4 and 4-BCD1) and the Program for Changjiang Scholars and Innovative Research Team in University (under grant-IRT_16R02). We thank professor Y.P. Wang and his team for allowing us to use the femtosecond laser micro-machining facility at Shenzhen University to fabricate microchannels on HCFs.

## Author contributions

The work was conducted at HK PolyU. P.Z. built the systems and conducted the experiments. H.B. and H.L.H. assisted in building the systems and conducting the experiments. P.Z. and Y.Z. performed numerical modeling. P.Z., Y.Z., W.J. and S.F. analyzed the results and prepared the manuscript. S.G., Y.W. and W.P. designed and fabricated the SR-AR-HCF. W.J. conceived the idea of SNR enhancement and developed the mathematical formulations. W.J. and S.F. coordinated the project.

## Competing interests

The authors declare no competing interests.
