## [Peer Review File · Nature Communications]

Reviewers' Comments:

Reviewer #1:

Remarks to the Author:

The Authors report on a gas sensor relying on a photothermal (PT) effect. The principal of PT gas detection relies on exciting the gas sample under test with a radiation source (preferably a laser) having an appropriate optical frequency (called pump/excitation laser) and subsequently measuring very subtle changes in the refractive index (RI) with an auxiliary laser (called probe laser), usually along the path-length of the excitation source. In this work Authors presented a sensor utilizing an anti-resonant hollow core fiber (AR-HCF) as a low-volume gas absorption cell. This approach is not new and was previously reported by several groups as it significantly simplifies the sensor layout, dramatically improves the pump-probe overlap and minimizes the gas volume required to perform the measurement. The AR-HCF used in the experiment had a low-loss window between 800 nm – 1700 nm, therefore the Authors decided to build the sensor based on widely available standard fiber components and target a gas having strong absorption band in that region – acetylene, a very standard gas for this wavelength region. Although the idea of using AR-HCFs in PT gas sensors is not new (especially in the 1.5 μm wavelength region), the Authors proposed a novel solution for probing the subtle PT-induced RI modulation – based on mode-phase difference. This required to excite a LP₁₁ mode structure in the HCF. Usually the PT-induced RI-modulation is probed in an interferometric configuration (e.g. Mach-Zehnder), where the reference arm is usually propagating outside the HCF, resulting in significant noise of the sensor. Here, the interferometer is relying on subtle phase fluctuations between modes propagating in the same optical fiber, thus common noise is rejected. This results in an outstanding SNR of ~ 14800 and a NEC of sub-ppt for an integration time of >60 minutes. The paper is written in a clear and comprehensive way. The results are mostly appropriately presented and discussed. The structure of the paper is following the standards of Nature Communications. References are well chosen and reflect most relevant experiments in the field of gas sensing in hollow core structures.

I was able to find only three grammatical mistakes in the paper:

1. Line 122-123; a verb is missing at the end of the sentence.
2. Line 172; the sentence should read "(...) repeatedly scanning (...)".
3. Line 206; "this leaves a room" should read "this leaves room".

Although I find the results exceptionally important for Researchers working in the field of gas sensors (particularly HCF-based sensors), the Authors should address the following suggestions:

1. Line 57-58; "seven to eight orders of magnitude" is not a precise statement.
2. Why the Authors have used 9.5 kHz as the sinewave modulation? How does the sensor respond to other modulation frequencies?
3. Line 151; Authors used a 1.95 bar overpressure to force the gas through the fiber. What was the time required for flushing the entire 4.7 meter-long fiber with the measured gas? Is the overpressure constant over the entire measurement process? How does the high pressure influence the line shape? What is the pressure distribution along the entire length of the fiber?
4. Line 157; Authors state the NEC for 1s averaging = 68 ppt based on the SNR measurement. Why is there a significant difference between the NEC values based on 1σ SNR and on the Allan-Werle deviation plot at $\Delta t = 1\text{s}$ in fig 4(c)?
5. The information about the sensitivity for >60 minutes averaging is in my opinion not valuable for comparison with other sensors previously published. It is common to present 1s and 100s values calculated from the Allan-Werle deviation plot (or SNR for 1σ). Averaging the signal for >60 min in any gas spectroscopy system is impractical, therefore the Authors should first and foremost state values for 1s and 100s averaging and subsequently mention in the text that due to superb common noise cancellation the system is capable of averaging longer, e.g. up to >60 min.
6. Line 178-186; In the first part of the paper Authors describe a sensor based on 4.7 meter-long AR-HCF. Yet, further on in the manuscript the Authors choose to describe a second sensor, which, although based on the same AR-HCF, has an entirely dissimilar layout. The differences are crucial: connections between the AR-HCF and the SMF, filling methods, shorter section of the AR-HCF. The results of both sensors are mentioned in the conclusion section of the paper although the measurements taken for the sensor based on the shorter AR-HCF are not as extensive as for the

first version of the sensor. Why is there no estimation of gas exchange in the setup using the longer fiber? Why the Authors omitted measuring parameters like Allan-Werle variation and long-term stability for the sensor based on a shorter AR-HCF?

7. Why the NEC calculated for the sorter version is drastically lower compared to the longer version? The difference in length is only $\times 6.4$ (0.73 m to 4.67 m), but the NEC difference is colossal – 392 times lower than for the longer version ($301 \text{ ppt}/0.766 \text{ ppt} = 392$). Is the lower SNR caused by the side-drilled channels? Personally, I find this confusing and I suggest the Authors to focus only on one of the sensor setups in the paper. This will require presenting measurements of the gas exchange time in the long AR-HCF.

8. Line 186; References are required.

9. Line 191; Why is the power from the probe beam so weak (80 μW)? The fiber does not introduce that much losses.

10. Line 209; I would be cautious with statements suggesting coupling optical powers of $> 1\text{W}$ into an AR-HCF, especially given the method used by the Authors (free space section and several dB of attenuation of the mechanical splice).

11. How was the 4.7 meter fiber secured during the measurements? Was it spooled on a fiber holder?

12. What was the reason the Authors did not try to manufacture filling holes along the 4.6 m fiber, like it was done for the sensor based on the shorter AR-HCF?

13. Was the AR-HCF home-manufactured or is it a commercial fiber? Was the fiber optimized during the simulation process to enable stable LP₁₁ mode transmission?

14. Is this sensor configuration reproducible in other wavelength regions? Is there a limitation to use this technique in combination with AR-HCFs having transmission windows in the mid-IR? Additionally, I am sceptic about the reproducibility of the presented experiments. This sensor relies on exciting an LP₁₁ mode in a AR-HCF and keeping this mode structure through the entire measurement period. As mentioned by the Authors, the fiber is capable of transmitting two orthogonal states of LP₁₁. This is in my opinion an additional drawback, as the pump excitation laser has to tightly match the optical path-length of the probe laser beam in order to observe reasonable PT-induced RI modulation effect. This is not trivial and in my opinion the Authors should put more effort into describing the technological process behind a non-complex and repeatable method for exciting the required mode structure in the fiber and maintaining it throughout the measurements.

Summarizing, I find this paper very interesting. The proposed sensing method is straightforward, achieves outstanding SNR and, as claimed by the Authors, is stable against environmental disturbances. If proven to be reproducible (based on the information provided in the manuscript I am personally skeptical), this method could branch-out into new interesting gas sensing methods. Nevertheless, some choices made by the Authors when composing the manuscript are questionable. Especially the choice to describe two significantly different sensor configurations, present different sets of measurements, and cherry-pick the measured performance parameters in the conclusion section. The Authors should in my opinion focus on one sensor layout and present an comprehensive set of measurements. Additionally, I think the manuscript would benefit if the Authors addressed the issues and suggestions listed above in the improved version of the manuscript.

Reviewer #2:

Remarks to the Author:

This submission describes an intriguing and, from this reviewer's perspective, entirely novel approach to in-fibre gas sensing based on the much explored hollow core concepts but using anti-resonant fibres which permit relatively low loss wide optical bandwidth transmission over useful possible absorption cell lengths up to a few or even a few tens of metres. The approach – namely using mode-mode interference between the two lowest order propagating modes – has demonstrated an unprecedented combination of sensitivity and dynamic range. The submission is certainly well worthy of publication, and whilst it could even be accepted in its present form it

could benefit from clarification of a few basic points in order to present a possible more balanced perspective on the prospects for this system. Here are just a few suggestions:

- The two modes shown in figure 1 b and 1 c do need a comment on polarisation and also on the second order mode possibility for horizontal alignment in addition to the vertical alignment shown. Also a comment on launch stability may also be useful – in the feed fibre (would that matter if the splice is sufficiently stable?) and on possible polarization impact in the feed and return single mode fibres.

- If no other modes are launched as the launch conditions vary (comment in fig 1 caption) then this needs an explanation in the light of the comments in the previous point. (Just a few words on both points will suffice – but the reader does need some 'comfort factor' here)

- Is there an assumption somewhere in here that the pump and probe beams will follow identical mode distributions between low and higher order spatial modes? This needs a brief comment since the discussion around figure 2 seems to imply similar paths. The modes will not perfectly overlap since the wavelengths are slightly different and also is there any optimum which best exploits the varying transverse locations of intensity maxima of the pump along the absorption cell due to mode:mode interference as the pump beam propagates?

- The first paragraph of the paragraph referring to figure 3 (line 132 of the pdf) requires some rephrasing. At present it reads as the absorption line peak being modulated at a frequency of... What is modulated? Presumably the pump intensity? Or is the pump being swept in wavelength – or in practice more likely -both?

- And in all cases, how is long term stability assured in terms of wavelength modulation depth and total pump intensity?

- The pump power corresponds to a very high power density in the fibre core. Some comments on safety may be useful – especially for flammable gases like acetylene. Detection in air is perhaps the more frequent situation, and so some comments on these power levels and corresponding densities are needed.

- The onset of non-linearity in the system response at around 1% concentration needs some critical analysis. A brief insight to why this happens coupled to some projections on how this non linearity may be accommodated in practice would be a very useful insight and would also indicate to the reader that the authors can justify any assertions that this can be usefully accommodated.

- The discussion sections also raise some queries. The limit on linearity of output signal vs pump power needs a bit more explanation. There are safety limits as mentioned but the implication is also there on keeping the pump power and wavelength very stable in practice. How?

- The input and output fibres are in effect the couples on the probe interferometer and on the pump source relative modal launch powers and phases, so they do play a critical part. The assumption here is that once the mechanics of the input and output splices is established, this must be stable and that the polarisation states of the input modes into the hollow core fibre don't drift in any way. Relative intermodal phase stability for both pump and probe within the hollow core fibre itself is also important. This may warrant some discussion.

- The comment on the 'non-linear transfer function beyond the weak absorption approximation' needs some elaboration. Relating this to the 50% absorption of the pump power also needs some explanation.

So overall, the technical content of the submission is most certainly of interest and the concept is certainly suitable for publication. From this reviewer's perspective the submission did however

need to clarify a number of discussion topics, of which the comments above are some examples. In this context, the authors have clearly established a 'language' which is clear within the collaboration, but for the external interested party (like this reviewer) the terminology needs just some simple clarification. With some attention to this, the submission could well become established as a landmark contribution in the field of fibre based gas sensing.

Reviewers' comments:

Reviewer #1 (Remarks to the Author):

The Authors report on a gas sensor relying on a photothermal (PT) effect. The principal of PT gas detection relies on exciting the gas sample under test with a radiation source (preferably a laser) having an appropriate optical frequency (called pump/excitation laser) and subsequently measuring very subtle changes in the refractive index (RI) with an axillary laser (called probe laser), usually along the path-length of the excitation source. In this work Authors presented a sensor utilizing an anti-resonant hollow core fiber (AR-HCF) as a low-volume gas absorption cell. This approach is not new and was previously reported by several groups as it significantly simplifies the sensor layout, dramatically improves the pump-probe overlap and minimizes the gas volume required to perform the measurement. The AR-HCF used in the experiment had a low-loss window between 800 nm – 1700 nm, therefore the Authors decided to build the sensor based on widely available standard fiber components and target a gas having strong absorption band in that region – acetylene, a very standard gas for this wavelength region. Although the idea of using AR-HCFs in PT gas sensors is not new (especially in the 1.5 μm wavelength region), the Authors proposed a novel solution for probing the subtle PT-induced RI modulation – based on mode-phase difference. This required to excite a LP₁₁ mode structure in the HCF. Usually the PT-induced RI-modulation is probed in an interferometric configuration (e.g. Mach-Zehnder), where the reference arm is usually propagating outside the HCF, resulting in significant noise of the sensor. Here, the interferometer is relying on subtle phase fluctuations between modes propagating in the same optical fiber, thus common noise is rejected. This results in an outstanding SNR of ~ 14800 and a NEC of sub-ppt for an integration time of >60 minutes. The paper is written in a clear and comprehensive way. The results are mostly appropriately presented and discussed. The structure of the paper is following the standards of Nature Communications. References are well chosen and reflect most relevant experiments in the field of gas sensing in hollow core structures.

I was able to find only three grammatical mistakes in the paper:

1. Line 122-123; a verb is missing at the end of the sentence.
2. Line 172; the sentence should read “(...) repeatedly scanning (...)”.
3. Line 206; “this leaves a room” should read “this leaves room”.

Although I find the results exceptionally important for Researchers working in the field of gas sensors (particularly HCF-based sensors), the Authors should address the following suggestions:

1. Line 57-58; “seven to eight orders of magnitude” is not a precise statement.
2. Why the Authors have used 9.5 kHz as the sinewave modulation? How does the sensor respond to other modulation frequencies?
3. Line 151; Authors used a 1.95 bar overpressure to force the gas through the fiber. What was the time required for flushing the entire 4.7 meter-long fiber with the measured gas? Is the overpressure constant over the entire measurement process? How does the high pressure influence the line shape? What is the pressure distribution along

the entire length of the fiber?

4. Line 157; Authors state the NEC for 1s averaging = 68 ppt based on the SNR measurement. Why is there a significant difference between the NEC values based on 1σ SNR and on the Allan-Werle deviation plot at $\Delta t = 1s$ in fig 4(c)?

5. The information about the sensitivity for >60 minutes averaging is in my opinion not valuable for comparison with other sensors previously published. It is common to present 1s and 100s values calculated from the Allan-Werle deviation plot (or SNR for 1σ). Averaging the signal for >60 min in any gas spectroscopy system is impractical, therefore the Authors should first and foremost state values for 1s and 100s averaging and subsequently mention in the text that due to superb common noise cancellation the system is capable of averaging longer, e.g. up to >60 min.

6. Line 178-186; In the first part of the paper Authors describe a sensor based on 4.7 meter-long AR-HCF. Yet, further on in the manuscript the Authors choose to describe a second sensor, which, although based on the same AR-HCF, has an entirely dissimilar layout. The differences are crucial: connections between the AR-HCF and the SMF, filling methods, shorter section of the AR-HCF. The results of both sensors are mentioned in the conclusion section of the paper although the measurements taken for the sensor based on the shorter AR-HCF are not as extensive as for the first version of the sensor. Why is there no estimation of gas exchange in the setup using the longer fiber? Why the Authors omitted measuring parameters like Allan-Werle variation and long-term stability for the sensor based on a shorter AR-HCF?

7. Why the NEC calculated for the shorter version is drastically lower compared to the longer version? The difference in length is only x6.4 (0.73 m to 4.67 m), but the NEC difference is colossal – 392 times lower than for the longer version (301 ppt/0.766 ppt = 392). Is the lower SNR caused by the side-drilled channels? Personally, I find this confusing and I suggest the Authors to focus only on one of the sensor setups in the paper. This will require presenting measurements of the gas exchange time in the long AR-HCF.

8. Line 186; References are required.

9. Line 191; Why is the power from the probe beam so weak (80 uW)? The fiber does not introduce that much losses.

10. Line 209; I would be cautious with statements suggesting coupling optical powers of > 1W into an AR-HCF, especially given the method used by the Authors (free space section and several dB of attenuation of the mechanical splice).

11. How was the 4.7 meter fiber secured during the measurements? Was it spooled on a fiber holder?

12. What was the reason the Authors did not try to manufacture filling holes along the 4.6 m fiber, like it was done for the sensor based on the shorter AR-HCF?

13. Was the AR-HCF home-manufactured or is it a commercial fiber? Was the fiber optimized during the simulation process to enable stable LP11 mode transmission?

14. Is this sensor configuration reproducible in other wavelength regions? Is there a limitation to use this technique in combination with AR-HCFs having transmission windows in the mid-IR?

Additionally, I am sceptic about the reproducibility of the presented experiments. This

sensor relies on exciting an LP₁₁ mode in a AR-HCF and keeping this mode structure through the entire measurement period. As mentioned by the Authors, the fiber is capable of transmitting two orthogonal states of LP₁₁. This is in my opinion an additional drawback, as the pump excitation laser has to tightly match the optical path-length of the probe laser beam in order to observe reasonable PT-induced RI modulation effect. This is not trivial and in my opinion the Authors should put more effort into describing the technological process behind an non-complex and repeatable method for exciting the required mode structure in the fiber and maintaining it throughout the measurements.

Summarizing, I find this paper very interesting. The proposed sensing method is straightforward, achieves outstanding SNR and, as claimed by the Authors, is stable against environmental disturbances. If proven to be reproducible (based on the information provided in the manuscript I am personally skeptical), this method could branch-out into new interesting gas sensing methods.

Nevertheless, some choices made by the Authors when composing the manuscript are questionable. Especially the choice to describe two significantly different sensor configurations, present different sets of measurements, and cherry-pick the measured performance parameters in the conclusion section. The Authors should in my opinion focus on one sensor layout and present an comprehensive set of measurements. Additionally, I think the manuscript would benefit if the Authors addressed the issues and suggestions listed above in the improved version of the manuscript.

Reviewer #2 (Remarks to the Author):

This submission describes an intriguing and, from this reviewer's perspective, entirely novel approach to in-fibre gas sensing based on the much explored hollow core concepts but using anti-resonant fibres which permit relatively low loss wide optical bandwidth transmission over useful possible absorption cell lengths up to a few or even a few tens of metres. The approach – namely using mode-mode interference between the two lowest order propagating modes – has demonstrated an unprecedented combination of sensitivity and dynamic range. The submission is certainly well worthy of publication, and whilst it could even be accepted in its present form it could benefit from clarification of a few basic points in order to present a possible more balanced perspective on the prospects for this system. Here are just a few suggestions:

- The two modes shown in figure 1 b and 1 c do need a comment on polarisation and also on the second order mode possibility for horizontal alignment in addition to the vertical alignment shown. Also a comment on launch stability may also be useful – in the feed fibre (would that matter if the splice is sufficiently stable?) and on possible polarization impact in the feed and return single mode fibres.

- If no other modes are launched as the launch conditions vary (comment in fig 1 caption) then this needs an explanation in the light of the comments in the previous point. (Just a few words on both points will suffice – but the reader does need some ‘comfort factor’ here)

- Is there an assumption somewhere in here that the pump and probe beams will follow identical mode distributions between low and higher order spatial modes? This needs a brief comment since the discussion around figure 2 seems to imply similar paths. The modes will not perfectly overlap since the wavelengths are slightly different and also is there any optimum which best exploits the varying transverse locations of intensity maxima of the pump along the absorption cell due to mode:mode interference as the pump beam propagates?

- The first paragraph of the paragraph referring to figure 3 (line 132 of the pdf) requires some rephrasing. At present it reads as the absorption line peak being modulated at a frequency of... What is modulated? Presumably the pump intensity? Or is the pump being swept in wavelength – or in practice more likely -both?

- And in all cases, how is long term stability assured in terms of wavelength modulation depth and total pump intensity?

- The pump power corresponds to a very high power density in the fibre core. Some comments on safety may be useful – especially for flammable gases like acetylene. Detection in air is perhaps the more frequent situation, and so some comments on these power levels and corresponding densities are needed.

- The onset of non-linearity in the system response at around 1% concentration needs some critical analysis. A brief insight to why this happens coupled to some projections on how this non linearity may be accommodated in practice would be a very useful insight and would also indicate to the reader that the authors can justify any assertions that this can be usefully accommodated.

- The discussion sections also raise some queries. The limit on linearity of output signal vs pump power needs a bit more explanation. There are safety limits as mentioned but the implication is also there on keeping the pump power and wavelength very stable in practice. How?

- The input and output fibres are in effect the couples on the probe interferometer and on the pump source relative modal launch powers and phases, so they do play a critical part. The assumption here is that once the mechanics of the input and output splices is established, this must be stable and that the polarisation states of the input modes into the hollow core fibre don't drift in any way. Relative intermodal phase stability for both pump and probe within the hollow core fibre itself is also important. This may warrant some discussion.

- The comment on the 'non-linear transfer function beyond the weak absorption approximation' needs some elaboration. Relating this to the 50% absorption of the pump power also needs some explanation.

So overall, the technical content of the submission is most certainly of interest and the concept is certainly suitable for publication. From this reviewer's perspective the submission did however need to clarify a number of discussion topics, of which the comments above are some examples. In this context, the authors have clearly established a 'language' which is clear within the collaboration, but for the external interested party (like this reviewer) the terminology needs just some simple clarification. With some attention to this, the submission could well become established as a landmark contribution in the field of fibre based gas sensing.

Response to reviewer 1 :

Comment 1.1

The Authors report on a gas sensor relying on a photothermal (PT) effect. The principal of PT gas detection relies on exciting the gas sample under test with a radiation source (preferably a laser) having an appropriate optical frequency (called pump/excitation laser) and subsequently measuring very subtle changes in the refractive index (RI) with an axillary laser (called probe laser), usually along the path-length of the excitation source. In this work Authors presented a sensor utilizing an anti-resonant hollow core fiber (AR-HCF) as a low-volume gas absorption cell. This approach is not new and was previously reported by several groups as it significantly simplifies the sensor layout, dramatically improves the pump-probe overlap and minimizes the gas volume required to perform the measurement. The AR-HCF used in the experiment had a low-loss window between 800 nm – 1700 nm, therefore the Authors decided to build the sensor based on widely available standard fiber components and target a gas having strong absorption band in that region – acetylene, a very standard gas for this wavelength region. Although the idea of using AR-HCFs in PT gas sensors is not new (especially in the 1.5 μm wavelength region), the Authors proposed a novel solution for probing the subtle PT-induced RI modulation – based on mode-phase difference. This required to excite a LP₁₁ mode structure in the HCF. Usually the PT-induced RI-modulation is probed in an interferometric configuration (e.g. Mach-Zehnder), where the reference arm is usually propagating outside the HCF, resulting in significant noise of the sensor. Here, the interferometer is relying on subtle phase fluctuations between modes propagating in the same optical fiber, thus common noise is rejected. This results in an outstanding SNR of ~ 14800 and a NEC of sub-ppt for an integration time of >60 minutes. The paper is written in a clear and comprehensive way. The results are mostly appropriately presented and discussed. The structure of the paper is following the standards of Nature Communications. References are well chosen and reflect most relevant experiments in the field of gas sensing in hollow core structures.

Response:

We thank the reviewer for the accurate comments.

Comment 1.2

I was able to find only three grammatical mistakes in the paper:

1. Line 122-123; a verb is missing at the end of the sentence.
2. Line 172; the sentence should read “() repeatedly scanning (...)”.
3. Line 206; “this leaves a room” should read “this leaves room”.

Response:

We have corrected these mistakes in the revised **Manuscript**.

Comment 1.3

Although I find the results exceptionally important for Researches working in the field of gas sensors (particularly HCF-based sensors), the Authors should address the following suggestions:

Line 57-58; “seven to eight orders of magnitude” is not a precise statement.

Response:

We have changed statement to ‘seven orders of magnitude’ in the revised Manuscript.

Comment 1.4

Why the Authors have used 9.5 kHz as the sinewave modulation? How does the sensor respond to other modulation frequencies?

Response:

We have numerically simulated the frequency response of PT phase modulation for MPD-PTS with the gas-filled SR-AR-HCF. The results are shown in Fig.S2.1 in **Supplementary 2**. For easy reference, the figure is also copied below. The PT modulation efficiency in terms of normalized modulation coefficient k^* is constant at low heat-modulation frequencies and decreases at higher frequencies. In our experiments, we choose 19 kHz, corresponding to about 88% the maximum value of k^* . In principle, we could operate at any heat-modulation frequency of below or around 19 kHz without compromising the PT phase modulation. However, the detection noise is of a $1/f$ dependence and hence we choose the pump wavelength modulation frequency to be 9.5 kHz, which corresponds to 19 kHz of the heat-modulation frequency (i.e., second harmonic of the pump modulation). Operating at around this frequency simultaneously achieves large PT phase modulation as well as lower detection noise, maximizing the signal to noise ratio.

Fig.S2.1. Modulation coefficient k^* as function of heat-modulation frequency that is the second harmonic frequency ($2f$) of pump modulation. All the results are obtained under the condition of ambient temperature with the pressure of 1.5 bar. The vertical dashed line indicated the operating frequency ($2f$) in our experiments.

The detailed mathematical formulation, numerical simulation and discussion can be found in **Supplementary 2**.

Comment 1.5

Authors used a 1.95 bar overpressure to force the gas through the fiber. What was the time required for flushing the entire 4.7-meter-long fiber with the measured gas? Is the overpressure constant over the entire measurement process? How does the high pressure influence the line shape? What is the pressure distribution along the entire length of the fiber?

Response:

It takes ~4 mins to fill the 4.67-meter-long SR-AR-HCF with measured gas by keeping a constant gas pressure of ~1.95 bar at the input end while output end kept open to atmospheric air. The measurement was conducted after the measured gas fills the entire length of the SR-AR-HCF and reaches a steady flow state. During the measurement, the measured gas flew through the SR-AR-HCF continuously and the gas pressure was kept at 1.95 bar at the input end and ~1 bar at the output end.

For acetylene in air, the lineshape could be approximated as a Lorentzian for gas pressure (P) from 1 to a few bars^[1]. For a pressure change from 1 to 1.95 bar, the center of the P(13) line of acetylene would shift ~ -209 MHz and linewidth increase from ~4.7 to ~12.9 GHz.

The gas pressure distribution along the fiber could be expressed as^[2]

$$P(z) = \sqrt{P_A^2 + \frac{z}{L}(P_B^2 - P_A^2)}$$

with

$$\begin{cases} P_A = 1.95 \text{ bar} \\ P_B = 1 \text{ bar} \\ L = 4.67 \text{ m} \end{cases}$$

where L is the length of fiber, $P_{A,B}$ the gas pressure at the ends of SR-AR-HCF (input end A and output end B), and z the position respect to input end A . The average pressure over the entire length of SR-AR-HCF could be calculated by

$$P_{avr} = 1/L \int_0^L P(z) dz \approx 1.5 \text{ bar}$$

The above discussion is included in **Supplementary 5 - *Test of dynamic range and response time.***

Comment 1.6

Authors state the NEC for 1s averaging = 68 ppt based on the SNR measurement. Why is there a significant difference between the NEC values based on 1σ SNR and on the Allan-Werle deviation plot at $\Delta t = 1$ s in fig 4(c)?

Response:

The stated 1σ NEC of 68 ppt was achieved with the SR830 lock-in amplifier with 1s lock-in time constant and 18 dB/oct roll-off, which corresponds a noise equivalent

detection bandwidth of ~ 0.094 Hz (Please see the handbook of the SR830 lock-in amplifier in detail). If we use the ideal low-pass filter model with sharp cutoff frequency, this corresponds to an equivalent integration time of ~ 10 s (instead of 1s). From the Allan-Werle deviation plot, 10s averaging time corresponds to 1σ NEC of ~ 43 ppt, which is not far away from ~ 68 ppt obtained from the lock-in measurement.

Please see the discussion in **Supplementary 5 - *Test of lower detection limit and long-term stability.***

Comment 1.7

The information about the sensitivity for >60 minutes averaging is in my opinion not valuable for comparison with other sensors previously published. It is common to present 1s and 100s values calculated from the Allan-Werle deviation plot (or SNR for 1σ). Averaging the signal for >60 min in any gas spectroscopy system is impractical, therefore the Authors should first and foremost state values for 1s and 100s averaging and subsequently mention in the text that due to superb common noise cancellation the system is capable of averaging longer, e.g. up to >60 min.

Response:

Thank you for the comments, which is very valuable. We have made changes in the revised **Manuscript**.

‘The 1σ NEC obtained from the Allan-Werle deviation plot is ~ 43 ppt and ~ 15 ppt respectively for 10 and 100 s averaging time. The system is capable of averaging over a much longer time of up to a few thousand seconds due to superb common mode noise cancellation...’ We listed the results of the 10s (instead of 1s) averaging time to make it easier to compare with the lock-in measurement.

Please see the paragraph just above Fig.4 in the revised **Manuscript**.

Comment 1.8

Line 178-186; In the first part of the paper Authors describe a sensor based on 4.7-meter-long AR-HCF. Yet, further on in the manuscript the Authors choose to describe a second sensor, which, although based on the same AR-HCF, has an entirely dissimilar layout. The differences are crucial: connections between the AR-HCF and the SMF, filling methods, shorter section of the AR-HCF. The results of both sensors are mentioned in the conclusion section of the paper although the measurements taken for the sensor based on the shorter AR-HCF are not as extensive as for the first version of the sensor. Why is there no estimation of gas exchange in the setup using the longer fiber? Why the Authors omitted measuring parameters like Allan-Werle variation and long-term stability for the sensor based on a shorter AR-HCF?

What was the reason the Authors did not try to manufacture filling holes along the 4.6 m fiber, like it was done for the sensor based on the shorter AR-HCF?

Response:

Thanks for the comments. We agree with the reviewer that it would be ideal to use

the 4.67-meter-long SR-AR-HCF sample to conduct all the experiments. However, it is practically not realistic for us to do so now with our current lab facilities. The femtosecond laser micro-machining system we used could only fabricate high quality filling-holes on a straight hollow-core fiber that is fixed on a plate before the start of hole-drilling. Previously, we made hundreds of holes along a single hollow-core photonic crystal fiber (HC-PCF) by fixing the HC-PCF on a plate with multiple straight sections with small bends between them. However, the state-of-the-art SR-AR-HCF cannot be bent to small diameters (the bending loss is $\sim 8\text{dB/m}$ for bend diameter smaller than $4\text{cm}^{[3]}$), hence the arrangement with multiple straight sections with a small bend in between (to connect the straight sections) cannot be achieved. In our experiment, we coiled the 4.67-meter-long SR-AR-HCF into loops of $\sim 18\text{cm}$ in diameter, we found it is extremely difficult to fabricate multiple high-quality filling holes on the curved fiber using our current setup. But we were able to drill multiple holes on a shorter length of straight SR-AR-HCF (i.e., $\sim 0.74\text{m}$) that was fixed to an aluminum plate, the SR-AR-HCF was mechanically spliced (permanently) to SMFs at both ends, which allows on-line monitoring the transmission loss during the fabrication of the holes. Drilling many holes on a curved fiber may be possible in future by using the more advanced micro-machining systems, which will be a future work.

Therefore, we decided to experimentally test the NEC and the long-term stability by pressuring gas into the 4.67-long SR-AR-HCF with pressure of $\sim 1.95\text{ bar}$ at the input end and output end open to atmospheric pressure. We used calibrated gas sample stored in a pressurized gas cylinder with known acetylene concentration for these experiments, and a complete filling of gas into the whole length of SR-AR-HCF took about 4 min. Without pressuring, it could take over 3 days by following the calculation according to reference [4].

However, for the measurement of dynamic range, we need to repeat experiment many times with the SR-AR-HCF filled with different gas concentrations, which are done by mixing calibrated gas samples with nitrogen by using flow meters in our lab. The gas mixtures are at $\sim 1\text{ bar}$ and we do not have the facility to pressurize the mixed gas sample to a higher-pressure level while still maintain high precision in concentration and pressure level. Hence, we decided to use the shorter sample ($\sim 0.74\text{m}$) of SR-AR-HCF with multiple filling holes to test the dynamic range as well as the response time. For the shorter SR-AR-HCF sample, the hollow-core can be filled via the filling-holes repeatedly, at around atmospheric pressure, with gas of different concentrations in a short time without the need to pressurize the gas.

Based on the reviewers' comments, we re-conducted experiments with a newly prepared 0.74-m-long-SR-AR-HCF gas cell, and the results are presented in the revised **Manuscript**. As suggested by this reviewer, we also conducted Allan-Werle derivation analysis, as shown in below (Fig.S5.1 in the **Supplementary 5**), which demonstrated NEC of $\sim 500\text{ ppt}$ with averaging time of $\sim 100\text{s}$. It could be averaged for a longer time (e.g., $>1000\text{ s}$) to achieve better NEC, indicating the system is stable over a longer term.

Fig.S5.1 Allan-Werle deviation analysis for the newly prepared 0.74-m-long SR-AR-HCF gas cell. It is based on the noise data over a period of 2 hours, which is shown in the inset. The gas pressure is ~ 1.5 bar. The lock-in time constant is 100 ms with 18 dB/oct roll-off, corresponding to 0.94 detection bandwidth. The probe power at the PD is about -9.5 dBm.

The above comments are included in the updated **Supplementary 5 - *Test of dynamic range and response time.***

Comment 1.9

Why the NEC calculated for the shorter version is drastically lower compared to the longer version? The difference in length is only $\times 6.4$ (0.73 m to 4.67 m), but the NEC difference is colossal – 392 times lower than for the longer version (301 ppt/0.766 ppt = 392). Is the lower SNR caused by the side-drilled channels? Personally, I find this confusing and I suggest the Authors to focus only on one of the sensor setups in the paper. This will require presenting measurements of the gas exchange time in the long AR-HCF.

Response:

The PT phase modulation is affected by the length of SR-AR-HCF, gas pressure and concentration, pump power level in the hollow-core as well as the fractional pump power η in the LP₀₁ mode (see Eqs. (S2.10) – (S2.13) in **Supplementary 2**). Since the PT phase modulation is detected by the dual-mode probe interferometer, the detected signal is also affected by the fringe contrast.

For the 4.67-m-long sample, the alignment between the output end of the SR-AR-HCF and the SMF can be adjusted by using the translation stages (see Fig.S4.1 in the **Supplementary 4**). This enables us to optimize the alignment at this joint to achieve the largest $2f$ signal from the lock-in. However, for the 0.73-m-long sample used in the original manuscript, the offset at the two joints between the SR-AR-HCF and the SMFs are difficult to optimize because of the limited degree of freedom offered by the mechanical splicing with ceramic ferrules and sleeve. The joints are permanent and could not be adjusted after they are fixed together by UV glue.

Instead of using the 0.73-m-long SR-AR-HCF sample in the original manuscript, we made a new 0.74-m-long-SR-AR-HCF with 34 microchannels (filling-holes) (see Fig.S4.3a in the **Supplementary 5**) to re-conduct the experiment and obtain more data.

We tested the NEC, dynamic range and performed Allan-Werle deviation analysis. The 0.74-m-long sample is still not optimized but has considerable better performance over the previous 0.73-m-long sample. Below is a comparison of the NEC between the 4.67-m-long and the 0.74-m-long sample with 100s averaging time.

The NEC of the 4.67-m-SR-AR-HCF is ~15ppt for averaging time of ~100s while it is ~500 ppt for the newly prepared 0.74-m-long SR-AR-HCF gas cell (the Allan deviation plot is shown in Fig. S5.1 in the **Supplementary 5** and also shown above), the ratio is $500/15=33$. Considering the length factor of $4.67/0.74 = 6.3$, the remaining $33/6.3=5.3$ times NEC deterioration is believed to be caused by the non-optimized alignment between the SR-AR-HCF and the pump input SMF, which could result in different pump power loss at the joint as well as different fractional pump power η in the LP₀₁ mode.

Please see discussion in updated **Supplementary 5 - *Test of dynamic range and response time***

Comment 1.10

Line 186; References are required.

Response:

We added a reference in the revised **Manuscript**.

Comment 1.11

Line 191; Why is the power from the probe beam so weak (80 uW)? The fiber does not introduce that much losses.

Response:

The relative low power is mainly due to the joint loss between the SR-AR-HCF and SMFs. There is a larger mode field mismatch, MFD~22 μ m for the LP₀₁ mode of the SR-AR-HCF while ~10.4 μ m for the SMF. To form the dual-mode interferometer, lateral offset between SMF and SR-HCF is needed to excite the two modes, the joint loss from SMF to SR-AR-HCF is typically small, but could go up to 15 dB from SR-AR-HCF to SMF, which is influenced by lateral offset. We made another 0.74-m-long SR-AR-HCF gas cell and reconducted the gas detection experiment. The probe power at photodetector is about -9.5 dBm, which is stated in caption of Fig.6 in the **Manuscript**.

Actually, we did not pay too much attention on the power of the probe beam since a few tens of microwatts probe power at the photodetector would be sufficient for this application.

Please see **Supplementary 4** and the caption of Fig.6 in the **Manuscript**.

Comment 1.12

Line 209; I would be cautious with statements suggesting coupling optical powers of > 1W into an AR-HCF, especially given the method used by the Authors (free space

section and several dB of attenuation of the mechanical splice).

Response:

Thank you for the comments. Yes, we agree with the reviewer that with the current setup, it is not realistic to deliver >1W CW light power into the SR-AR-HCF, considering the gap, the joint loss, and the theoretical damage threshold of commercial SMF-28 used in this research (< 1W, as from Thorlabs website). What we mean here is that the SR-AR-HCF itself could handle a much higher optical power level and if there is a way to deliver the pump into the SR-AR-HCF by, for example, using a large mode area fiber without a gap in the joint or other alternatives, it could be possible to deliver more power into the SR-AR-HCF.

We have made changes in the *Discussion* section of the revised **Manuscript**.

Comment 1.13

How was the 4.7 meter fiber secured during the measurements? Was it spooled on a fiber holder?

Response:

The 4.67-m-long SR-AR-HCF was looped to a diameter of ~18 cm, and secured on an acrylic plate by sellotape.

The newly prepared 0.74-m-long SR-AR-HCF was kept straight, secured onto a 1-m-long narrow aluminum plate by sellotape, and put inside an acrylic tube as the gas chamber.

Please see the revision made in **Supplementary 4**.

Comment 1.14

What was the reason the Authors did not try to manufacture filling holes along the 4.6 m fiber, like it was done for the sensor based on the shorter AR-HCF?

Response:

Please see response to **Comment 1.8**.

Comment 1.15

Was the AR-HCF home-manufactured or is it a commercial fiber? Was the fiber optimized during the simulation process to enable stable LP₁₁ mode transmission?

Response:

SR-AR-HCF was home-made by the co-authors Shoufei Gao, Yingying Wang and Pu Wang. The fiber is not optimized for this application, but we found it supports two and only two groups of LP modes (i.e., LP₀₁ and LP₁₁) with stable mode patterns. We hope to optimize the fiber in future so that it could only support LP₀₁ and LP_{11a}, for example. After completion of this work, we noticed a recently published paper [5],

which states that stable guidance of preferred modes could be achieved relatively easily by altering the position of the silica capillary rings.

The above information is included in **Supplementary 4 - *The SR-AR-HCF***.

Comment 1.16

Is this sensor configuration reproducible in other wavelength regions? Is there a limitation to use this technique in combination with AR-HCFs having transmission windows in the mid-IR?

Response:

We believe that the configuration can be used for other wavelengths. In fact, with a similar configuration, we have successfully conducted experiments with a 1651nm pump and a 1550 nm probe for CH₄ detection.

We believe the method can also be used for other wavelength (e.g., mid IR) as long as it's within transmission band of SR-AR-HCF. We are currently working on a system with a mid-IR pump and a 1550 nm probe. The principle is the same with a slightly re-arrangement of the detection system.

Please see the comments in the ***Discussion*** section in the revised **Manuscript**.

Comment 1.17

Additionally, I am sceptic about the reproducibility of the presented experiments. This sensor relies on exciting an LP₁₁ mode in an AR-HCF and keeping this mode structure through the entire measurement period. As mentioned by the Authors, the fiber is capable of transmitting two orthogonal states of LP₁₁. This is not trivial and in my opinion the Authors should put more effort into describing the technological process behind a non-complex and repeatable method for exciting the required mode structure in the fiber and maintaining it throughout the measurements.

Summarizing, I find this paper very interesting. The proposed sensing method is straightforward, achieves outstanding SNR and, as claimed by the Authors, is stable against environmental disturbances. If proven to be reproducible (based on the information provided in the manuscript I am personally skeptical), this method could branch-out into new interesting gas sensing methods.

Response:

Stability of the system

The mode structure of the SR-AR-HCF is very stable in the lab environment. We have tested samples with SMF input to the SR-AR-HCF and observed the following:

1. By varying the launching condition, i.e., lateral offset between SMF and SR-AR-HCF, we can excite LP₀₁ mode in combination with one of the LP₁₁ mode (i.e., either LP_{11a} or LP_{11b}). The two-lobe pattern of the excited LP_{11a} or LP_{11b} mode does not vary with launch condition, although the intensity level may change. No other higher order mode was observed. (Please see caption of Fig.1 in the **Manuscript** as well as **Supplementary 4**)

2. We find that once the SMF/SR-AR-HCF joint is secured, perturbation on the SMF does not affect the lobe-orientation. Small perturbation on the SR-AR-HCF also does not affect the lobe-orientation; however, when a tight bend (of a few cm) is applied to the SR-AR-HCF, we observed conversion from LP_{11a} to LP_{11b} or vice versa. In our experiments, strong bend was avoided and hence the mode pattern does not change during the entire measurement process. This may be due to the considerably large index difference between LP_{11a} and LP_{11b} ($\sim 7 \times 10^{-5}$) and the fact the bending doesn't introduce large modal-birefringence due to most of the mode energy is in air and hence the mode birefringence is relatively insensitive to bending. However, as explained later in the response to **Comment 2.11**, changing input polarization state by the polarization controller (PC) does change the detected PT signal considerably. During our experiment, PC was firstly adjusted and then fixed. All other fiber components and fiber sections were secured by sellotapes. Stable operation can be maintained for many hours. (Please see **Supplementary 4**)

For future practical applications, polarization maintaining SMFs in combination with highly birefringent SR-AR-HCF would overcome problem of polarization. (Please see **discussion** part of the revised **Manuscript**).

3. As shown in Fig. 5 in the revised **Manuscript**, the system is stable over a long period of time. If the mode structure were not maintained throughout the experiment, the signal fluctuation would be much larger.

Which mode is launched, LP_{11a} or LP_{11b}?

In our experimental setup, the pump and the probe are launched from two ends of the SR-AR-HCF, hence we could not know if the pump and the probe are in the same LP₁₁ mode (e.g., both pump and probe are in LP_{11a}, or one in LP_{11a} and the other in LP_{11b}). But we know that no matter what combination, once it is launched, the combination will be fixed and maintained during the entire measurement process.

To examine the effect of different combination of the pump and probe modes on the differential PT phase modulation, we numerically calculated the differential PT phase modulation coefficient for various pump and probe mode combinations for varying fractional pump power η in the LP₀₁ mode. The results are shown in the **Figure S2.2** in the **Supplementary 2**, which is also copied below. According to the simulation results, for $\eta > 80\%$ (which is the case for the two samples used in the experiments), the modulation coefficient k^* is within 20% of the maximum value (i.e., the value for $\eta = 100\%$). This means that it is not a big issue whether the pump and the probe are launched into the same LP₁₁ mode or not, as long as a large percentage of the pump power is in the LP₀₁ mode. **Fig.S2.2** also shows that it is better to launch the pump and the probe into orthogonal LP₁₁ mode (i.e., one in LP_{11a} and the other in LP_{11b}) because the dependence on η is much weaker! The best choice is of course to launch all the pump power into the LP₀₁ mode (i.e., $\eta = 100\%$), where the k^* value is always maximized and around 1.6×10^{-7} rad cm ppm⁻¹m⁻¹mW⁻¹. This could in principle be done by launching the pump into LP₀₁ mode (via perfect alignment) and then using a long period grating inscribed along the SR-AR-HCF to resonantly couple part of the probe (not the pump) to LP_{11a} (or LP_{11b}) to form an in-line probe dual-mode interferometer

while leaving the pump beam unaffected.

Fig.S2.2 Modulation coefficient k^* as function of η for different mode combinations for the pump and probe beam. The results are obtained under the condition of ambient temperature and 1.5 bar gas pressure.

The above results and discussions are included in **Supplementary 2** and **Supplementary 4 - *The SR-AR-HCF***.

Comment 1.18

Nevertheless, some choices made by the Authors when composing the manuscript are questionable. Especially the choice to describe two significantly different sensor configurations, present different sets of measurements, and cherry-pick the measured performance parameters in the conclusion section. The Authors should in my opinion focus on one sensor layout and present a comprehensive set of measurements. Additionally, I think the manuscript would benefit if the Authors addressed the issues and suggestions listed above in the improved version of the manuscript.

Response:

Due to difficulties mentioned earlier, we could not test all the performance indicators with a single long SR-AR-HCF using our current facilities. The shorter sample was made to test the dynamic range and the potential for faster gas filling. We conducted more experiments with a newly prepared shorter sample and the results are included in the **Manuscript**.

We have revised the *Abstract* and the *Discussion* section in the **Manuscript** to clearly state that the results are from the two different samples.

Response to reviewer 2:

Comment 2.1

This submission describes an intriguing and, from this reviewer's perspective, entirely novel approach to in-fibre gas sensing based on the much explored hollow core concepts but using anti-resonant fibres which permit relatively low loss wide optical

bandwidth transmission over useful possible absorption cell lengths up to a few or even a few tens of metres. The approach – namely using mode-mode interference between the two lowest order propagating modes – has demonstrated an unprecedented combination of sensitivity and dynamic range. The submission is certainly well worthy of publication, and whilst it could even be accepted in its present form it could benefit from clarification of a few basic points in order to present a possible more balanced perspective on the prospects for this system. Here are just a few suggestions:

Response:

We would like to thank the reviewer for the positive comments.

Comment 2.2

The two modes shown in figure 1 b and 1 c do need a comment on polarisation and also on the second order mode possibility for horizontal alignment in addition to the vertical alignment shown.

Response:

Thank you for the kind suggestion. In the **caption of figure 1 and Supplementary 1** of the revised **Manuscript**, we have stated that there are two LP₁₁ mode (LP_{11a} and LP_{11b}), and we only drew one of them, and each mode also has orthogonal polarizations states.

Comment 2.3

a comment on launch stability may also be useful – in the feed fibre (would that matter if the splice is sufficiently stable?) and on possible polarization impact in the feed and return single mode fibres.

Response:

Once the splice is fixed and stable, the lobe-pattern of the mode will not change. However, varying input polarization states does cause variation in the detected PT signal. In our experiments, the PC and all the fiber components are secured by sellotape and the system is found very stable.

Please see response to **Comment 1.17** and **Comment 2.11**.

Comment 2.4

if no other modes are launched as the launch conditions vary (comment in fig 1 caption) then this needs an explanation in the light of the comments in the previous point. (Just a few words on both points will suffice – but the reader does need some ‘comfort factor’ here)

Response:

Thank you for the suggestion, we added a sentence in the **description of Figure 1** in

the revised **Manuscript**. Apart from the LP_{11} modes, no other higher-order modes were experimentally observed when input launching condition is varied. In effect, the SR-AR-HCF could only support the LP_{01} and LP_{11} (including LP_{11a} and LP_{11b}) modes.

It is theoretically possible for the fiber to support LP_{01} and one of the LP_{11} modes (e.g., LP_{11b}). However, such a fiber is currently not available and may be fabricated in future. Please see response to **Comment 1.15** and **Comment 1.17**.

Comment 2.5

Is there an assumption somewhere in here that the pump and probe beams will follow identical mode distributions between low and higher order spatial modes? This needs a brief comment since the discussion around figure 2 seems to imply similar paths. The modes will not perfectly overlap since the wavelengths are slightly different and also is there any optimum which best exploits the varying transverse locations of intensity maxima of the pump along the absorption cell due to mode:mode interference as the pump beam propagates?

Response:

In the **Manuscript**, heating profile (Fig.2c) is derived from absorption of pump beam shown in Fig.2b, hence they look similar. In Fig.2c, for the purpose of illustration, we only drew one of the probe LP_{11} mode (i.e., LP_{11b}). In fact, we did not make the assumption that pump and probe beams share the same propagating path in SR-AR-HCF. As stated in respond to **Comment 1.17**, the pump and probe beam could have four different combinations (i.e., pump: $LP_{01}+LP_{11a}/LP_{11b}$ and probe: $LP_{01}+LP_{11a}/LP_{11b}$) as shown in **Fig.S2.2** in the **Supplementary 2**.

The differential PT phase modulation due to the mode-mode interference term accumulated over 570 periods ($\sim 0.74m$) along the SR-AR-HCF approaches zero, and hence the magnitude of differential probe phase modulation only depends on the pump power level and fractional power ratio η , please see the discussion in **Supplementary 2**.

For the current system, the pump and probe wavelengths are respectively 1532nm and 1550 nm, the mode field distributions can be regarded as the same. We numerically calculated the PT phase modulation coefficient by using the exact mode fields at two wavelengths as well as the coefficient by assuming perfect mode field overlap, the difference is less than 1%.

Theoretically, differential PT phase modulation is maximized when the pump beam is completely in the LP_{01} mode. For different η and different combinations of pump and probe modes, the normalized differential PT phase modulation is shown in **Fig.S2.2** in response to **Comment 1.17**.

Comment 2.6

The first paragraph of the paragraph referring to figure 3 (line 132 of the pdf) requires some rephrasing. At present it reads as the absorption line peak being modulated at a frequency of... What is modulated? Presumably the pump intensity? Or is the pump

being swept in wavelength – or in practice more likely -both?

Response:

We rephrased that sentence following your comment in the revised **Manuscript**.
‘ The pump source is a 1.53- μm distributed feedback (DFB) laser and its wavelength is modulated sinusoidally at ~ 9.5 kHz, and at the same time scanned slowly across the P(13) absorption line of acetylene at 1532.830 nm’.

Comment 2.7

And in all cases, how is long term stability assured in terms of wavelength modulation depth and total pump intensity?

Response:

Thanks for the comments. We used commercial laser current (Thorlabs, LDC205C) and temperature (Thorlabs, TED200C) controllers in combination with the internal signal generator of Lock-in amplifier (Stanford Research Systems, SR830) to drive the pump source (Distributed-feedback laser[Eudyna, FLD5F15CX]) and perform modulation and an EDFA (Amonics, AEDFA-EX) for power amplification. The probe source we used is an external-cavity diode laser (Agilent 81600B), and signal is detected by a photodetector (Nirvana-2017).

According to long-term stability measurement result shown in Fig.5, the system is quite stable. However, we didn’t test separately the long-term stability of pump power and wavelength modulation depth, and the current local situation does not allow us to do so now.

We have included the equipment information in updated **Supplementary 5**.

Comment 2.8

The pump power corresponds to a very high-power density in the fibre core. Some comments on safety may be useful – especially for flammable gases like acetylene. Detection in air is perhaps the more frequent situation, and so some comments on these power levels and corresponding densities are needed.

Response:

For 1W pump power launched into SR-AR-HCF filled with 5% C_2H_2 at ambient condition, the highest temperature rise is calculated to be ~ 14 $^\circ\text{C}$, which is well below acetylene’s autoignition temperature of 305°C according to DIN EN 14522-2005. The pump power density for 1W in the hollow-core is $\sim 10^5$ W/cm^2 , far below breakdown value $\sim 10^{12}$ W/cm^2 of air, which is similar to that of C_2H_2 ^[6].

Safety problem is commented in the *Discussion* section of the revised **Manuscript**.

Comment 2.9

The onset of non-linearity in the system response at around 1% concentration needs

some critical analysis. A brief insight to why this happens coupled to some projections on how this nonlinearity may be accommodated in practice would be a very useful insight and would also indicate to the reader that the authors can justify any assertions that this can be usefully accommodated.

Response:

The non-linearity of the detection system is affected by two factors: the non-linear characteristic of the Beer-Lambert law and the non-linear (cosine) transfer function of interferometric detection.

According to the Beer-Lambert law, the differential PT phase modulation can be expressed as

$$\delta\phi = k*(1-e^{-\alpha CL})P_{\text{pump}} \approx k*\alpha CLP_{\text{pump}}$$

The above approximation is accurate in the weak absorption limit, i.e., $\alpha CL \ll 1$.

The relative error from linear approximation is

$$\epsilon_1 = \frac{(1 - e^{-\alpha CL}) - \alpha CL}{\alpha CL} \times 100\%$$

Obviously ϵ_1 will become larger for increasing αCL . For the P(13) line of acetylene with $\alpha = 1.0513 \text{ cm}^{-1}$, the concentration $C= 1\%$ and the SR-AR-HCF sample of $L=0.74 \text{ m}$, $1 - e^{-\alpha CL} \approx 50\%$ meaning about half of the pump power is absorbed by the gas, the relative error ϵ_1 is $\sim 30\%$.

Around the quadrature point, the probe intensity modulation at the output of the dual-mode interferometer is related to the differential phase modulation by

$$I_{\text{signal}} = A \sin(\delta\phi) \approx A\delta\phi \propto C$$

where A is a constant related to the probe power level and the fringe contrast. The approximation is accurate for $\delta\phi \ll 1$ because of $\sin(\delta\phi) \approx \delta\phi$. The relative error from linear approximation is

$$\epsilon_2 = \frac{\sin(\delta\phi) - \delta\phi}{\delta\phi} \times 100\%$$

For larger $\delta\phi$, the relative error ϵ_2 will become larger. With the same parameters above and for 110-mW pump power, the differential PT phase modulation is

$$\delta\phi = k*\alpha CLP_{\text{pump}} = 1.6 \times 10^{-7} \times 1.0513 \times \left(\frac{1\%}{1\text{ppm}}\right) \times 0.74 \times 110 = 0.137 \text{ rad}$$

The relative error ϵ_2 is $\sim 0.3\%$, well below the relative error ϵ_1 . And hence it does not need to be considered here.

The above discussion is included in **Supplementary 5**.

Comment 2.10

The discussion sections also raise some queries. The limit on linearity of output signal vs pump power needs a bit more explanation. There are safety limits as mentioned but the implication is also there on keeping the pump power and wavelength very stable in practice. How?

Response:

Please see the response to **Comment 2.7** (pump stability) & **2.8** (safety) & **2.9** (nonlinearity).

Comment 2.11

The input and output fibres are in effect the couples on the probe interferometer and on the pump source relative modal launch powers and phases, so they do play a critical part. The assumption here is that once the mechanics of the input and output splices is established, this must be stable and that the polarisation states of the input modes into the hollow core fibre don't drift in any way. Relative intermodal phase stability for both pump and probe within the hollow core fibre itself is also important. This may warrant some discussion.

Response:

Once the of the input and output are fixed and kept stable, mode pattern will be stable, as discussed in response to **Comment 1.17**.

According to our experimental observation, the polarization of probe laser does affect the PT signal. However, when the PC and other fiber components are secured by sellotape, the PT signal is found stable as observed in the experiment.

About intermodal phase stability:

The pump is a heat source. In the current setup, the two pump modes are launched into the SR-AR-HCF and they will interfere to cause the pump intensity distribution to vary along the SR-AR-HCF. This in turn will change the heating profile along the SR-AR-HCF. However, as analyzed in **Supplementary 2**. The change of pump intensity along the fiber due to mode-mode interference has negligible effect on the probe phase modulation, which depends primarily on the power ratio of the pump LP₀₁ and LP₁₁ mode, not the relative mode phase.

For the probe light, the relative phase between the probe modes could be affected by launch conditions (e.g., offset and polarization), which may change the operating point of the probe interferometer. However, in our experiments, the joints and fibers are fixed and stable and the system stability was quite good, as shown in the measurement results in Figs.4 and 5 in the **Manuscript**.

All these problems may be overcome by using polarization maintaining SMFs (commercially available) and high birefringent SR-AR-HCF (under research, not available yet), and the polarization maintaining system would be insensitive to polarization fluctuation and may be implemented in future real time systems.

Comments on the polarization and mode stability are included in **Discussion** section of revised **Manuscript**.

Comment 2.12

The comment on the 'non-linear transfer function beyond the weak absorption

approximation' needs some elaboration. Relating this to the **50%** absorption of the pump power also needs some explanation.

Response:

Please see the response to **Comment 2.9**.

Comment 2.13

So overall, the technical content of the submission is most certainly of interest and the concept is certainly suitable for publication. From this reviewer's perspective the submission did however need to clarify a number of discussion topics, of which the comments above are some examples. In this context, the authors have clearly established a 'language' which is clear within the collaboration, but for the external interested party (like this reviewer) the terminology needs just some simple clarification. With some attention to this, the submission could well become established as a landmark contribution in the field of fibre based gas sensing.

Response:

Thank you for the useful and critical comments.

Reference

- [1] Swann, W. C. & Gilbert, S. L. Pressure-induced shift and broadening of 1510–1540-nm acetylene wavelength calibration lines. *J. Opt. Soc. Am. B* **17**, 1263-1270 (2000).
- [2] Suda, A. et al. Generation of sub-10-fs, 5-mJ-optical pulses using a hollow fiber with a pressure gradient. *Appl. Phys. Lett.* **86**, 111-116 (2005).
- [3] Shoufei G , Yingying W , Xiaolu L , et al. Bending loss characterization in nodeless hollow-core anti-resonant fiber. *Opt. express* **24**, 14801-14811 (2016).
- [4] Hoo Y. L. et al. Design and modeling of a photonic crystal fiber gas sensor. *Appl. Optics* **42**, 3509-3515 (2003).
- [5] Osório, J. H. et al. Tailoring modal properties of inhibited-coupling guiding fibers by cladding modification. *Sci. Rep.* **9**, 1376 (2019).
- [6] Miziolek, A. W. & Sausa, R. C. Photochemical Ignition Studies. I. Laser Ignition of Flowing Premixed Gases. Technical Report BRL-TR-2644 (US Army Ballistic Research Laboratory, 1985).

Reviewers' Comments:

Reviewer #1:

Remarks to the Author:

The Authors have addressed the suggestions and questions raised by the Reviewers. The answers provided in the Authors Response file are in my opinion sufficient. I find the edited version of the manuscript significantly improved and thus I can now recommend the article to be published in NC without further changes.

Reviewer #2:

Remarks to the Author:

The authors have completed a conscientious and thorough response to the previous reviewers' comments. However, from this reviewer's perspective, there do remain a few, primarily editorial / clarification issues which could significantly enhance the current version. These clarifications include:

- In the abstract it could be helpful to quantify ultra-high sensitivity etc and also to mention that these results are for acetylene – a very co-operative sample material!
- End of para 2 of the introduction (line 49) gas consumption? Does this really imply small sample volume?
- Mention the length of fibre in the next paragraph (around line 55)
- Line 73 – 'refractive index' maybe should be 'effective index'?
- Just above equation 2 (line 114) there is mention of 'common mode rejection'. This needs rather more thorough explanation since inter-mode dispersion will be affected by temperature and strain changes = albeit small. But this photo-thermal effect is small too since both modes are travelling the same paths as they would if the photo thermal effect weren't there. The whole process relies on simply the varying interference sums between the two modes producing hotter or colder spots. The dynamic filtering (modulated pump) will definitely help. Please spend a little more time explaining the justifications for the 'common mode' assertion. This also includes the 'sensitivity ratio' discussion.
- On line 138, the comment on tuning the probe to the quadrature point' need clarification. How? And how is this kept stable?
- Figure 3 doesn't apparently show the sinusoidal modulation source
- On line 158 – are you really confident to 5 significant figures on SNR?
- Line 163 mentions 'superb' common mode rejection. What is 'superb'? Also fig 4 c raises some questions especially for integration times over 100seconds. (The roll off increases and the variability is pronounced – why?)
- Also – detection signal amplitude depends on probe and pump power levels, not to mention retaining the quadrature point. How stable are these over longer periods? This leads into figure 5 – which needs clear straightforward explanation of what these results actually mean and why.

One final comment – the story could be significantly improved by a short, but convincing, exploration of longer term stability issues. These are mentioned but pump and probe powers and,

for the probe, wavelength, are important. Also – whilst there are many comments on 'common mode rejection' there is no real evidence. What if the temperature of the cell is changed by, say, 50C?

These modifications could, for this reviewer, certainly make the submission more accessible to potential readers. Whilst many of these potential ambiguities could in principle be solved by sheer reader determination, it is definitely better to try to avoid this requirement. The concept overall is certainly interesting and worthy of publication..

Reviewers' comments:

Reviewer #1 (Remarks to the Author):

The Authors have addressed the suggestions and questions raised by the Reviewers. The answers provided in the Authors Response file are in my opinion sufficient. I find the edited version of the manuscript significantly improved and thus I can now recommend the article to be published in NC without further changes.

Sincerely,
Karol Krzempek

Reviewer #2 (Remarks to the Author):

The authors have completed a conscientious and thorough response to the previous reviewers' comments. However, from this reviewer's perspective, there do remain a few, primarily editorial / clarification issues which could significantly enhance the current version. These clarifications include:

- In the abstract it could be helpful to quantify ultra-high sensitivity etc and also to mention that these results are for acetylene – a very co-operative sample material!
- End of para 2 of the introduction (line 49) gas consumption? Does this really imply small sample volume?
- Mention the length of fibre in the next paragraph (around line 55)
- Line 73 – 'refractive index' maybe should be 'effective index'?
- Just above equation 2 (line 114) there is mention of 'common mode rejection'. This needs rather more thorough explanation since inter-mode dispersion will be affected by temperature and strain changes = albeit small. But this photo-thermal effect is small too since both modes are travelling the same paths as they would if the photo thermal effect weren't there. The whole process relies on simply the varying interference sums between the two modes producing hotter or colder spots. The dynamic filtering (modulated pump) will definitely help. Please spend a little more time explaining the justifications for the 'common mode' assertion. This also includes the 'sensitivity ratio' discussion.
- On line 138, the comment on tuning the probe to the quadrature point' need clarification. How? And how is this kept stable?
- Figure 3 doesn't apparently show the sinusoidal modulation source

- On line 158 – are you really confident to 5 significant figures on SNR?

- Line 163 mentions 'superb' common mode rejection. What is 'superb'? Also fig 4 c raises some questions especially for integration times over 100seconds. (The roll off increases and the variability is pronounced – why?)

- Also – detection signal amplitude depends on probe and pump power levels, not to mention retaining the quadrature point. How stable are these over longer periods? This leads into figure 5 – which needs clear straightforward explanation of what these results actually mean and why.

One final comment – the story could be significantly improved by a short, but convincing, exploration of longer term stability issues. These are mentioned but pump and probe powers and, for the probe, wavelength, are important. Also – whilst there are many comments on 'common mode rejection' there is no real evidence. What if the temperature of the cell is changed by, say, 50C?

These modifications could, for this reviewer, certainly make the submission more accessible to potential readers. Whilst many of these potential ambiguities could in principle be solved by sheer reader determination, it is definitely better to try to avoid this requirement. The concept overall is certainly interesting and worthy of publication.

Response to reviewer 2:

Comment 2.1

The authors have completed a conscientious and thorough response to the previous reviewers' comments. However, from this reviewer's perspective, there do remain a few, primarily editorial / clarification issues which could significantly enhance the current version. These clarifications include:

- In the abstract it could be helpful to quantify ultra-high sensitivity etc and also to mention that these results are for acetylene – a very co-operative sample material!

Response:

We added the quantified detection sensitivity and stability, and mentioned that the detected gas is acetylene in the *Abstract* of revised **Manuscript**.

Comment 2.2

- End of para 2 of the introduction (line 49) gas consumption? Does this really imply small sample volume?

Response:

Reduced gas consumption refers to that the gas consumption required for the SR-AR-HCF gas cell is much smaller than the normal bulky open-path cells. Operating in the NIR does not necessarily mean reduced gas consumption. We have made revision at the end of para 2 in the *Introduction* of the revised **Manuscript**.

Comment 2.3

- Mention the length of fibre in the next paragraph (around line 55)

Response:

In the revised **Manuscript**, we have mentioned the lengths of the SR-AR-HCF gas cells used.

Comment 2.4

- Line 73 – 'refractive index' maybe should be 'effective index'?

Response:

We thank the reviewer for pointing out this. We corrected it in the revised **Manuscript**.

Comment 2.5

- Just above equation 2 (line 114) there is mention of 'common mode rejection'. This

needs rather more thorough explanation since inter-mode dispersion will be affected by temperature and strain changes = albeit small. But this photo-thermal effect is small too since both modes are travelling the same paths as they would if the photo thermal effect weren't there. The whole process relies on simply the varying interference sums between the two modes producing hotter or colder spots. The dynamic filtering (modulated pump) will definitely help. Please spend a little more time explaining the justifications for the 'common mode' assertion. This also includes the 'sensitivity ratio' discussion.

Response:

The periodic pump modulation distinguishes PT phase modulation of the fundamental mode from external perturbation (such as environmental temperature and pressure), which enabled high sensitivity of gas absorption that has been described in previous works (see ref. [12, 13] in the manuscript)

What we mean here is that the sensitivity of the MPD to external perturbation is significantly reduced as compared with the sensitivity of the fundamental mode phase to the external perturbation. As stated in the paragraph after Equ.(3) of the **Manuscript** the reduction factor or the 'sensitivity ratio' is ~280 for temperature and ~170 for pressure. The reasons for this significant reduction of sensitivity has been explained in the paragraph after Fig.(2) of the **Manuscript** as well as in **Supplementary Note 2**. The larger the environmental perturbation, the bigger the phase change of the fundamental mode as well as the MPD, but the change in the MPD should be much smaller than the fundamental mode phase. Our experiments under ambient condition indeed showed that the detection limit and the stability are much better than those in the previous reports based on the detection of the fundamental mode phase. However, we did not conduct experiment over large temperature excursion of say 50°C, which will be a future work.

As demonstrated in **Supplementary Note 2**, the PT modulation in the MPD is primarily determined by the fractional pump powers in the two modes, which is not much affected by the interference term. It is maximized when all the pump power is in the LP₀₁ mode in the HCF (**Supplementary Fig. 3**). In our experiments, it would be better to launch as much pump power into the LP₀₁ mode as possible (i.e., minimal pump power into the LP₁₁ mode).

To avoid confusion, we deleted "common mode rejection" and made revisions in *Theory* and para 2 of Discussion in the revised **Manuscript**.

Comment 2.6

- On line 138, the comment on tuning the probe to the quadrature point' need clarification. How? And how is this kept stable?

Response:

In the present work, we simply tuned the probe wavelength, at the beginning of each experiment, to a quadrature point of dual-mode interference fringe, as mentioned in para 2 of Discussion in the **Manuscript**. The system was very stable, and the operating

point remained almost unchanged over the entire measurement period in the lab environment. We didn't use any feedback controller to stabilize the operating point. Instead we detected the $2f$ signal and the DC-component from the photodetector (not shown in Fig.3) and normalized the $2f$ signal by the DC-component to obtain the results shown in Fig. 5(c). For real time applications, stable operation may be achieved by servo controlling the probe wavelength to track the fringe quadrature.

Clarification of the quadrature operation has been included in the **Supplementary Note 6- *Test of lower detection limit and long-term stability.***

Comment 2.7

- Figure 3 doesn't apparently show the sinusoidal modulation source?

Response:

The sinusoidal modulation signal is from internal signal generator of LIA, which is stated in the revised **Manuscript.**

Comment 2.8

- On line 158 – are you really confident to 5 significant figures on SNR?

Response:

The SNR was obtained through dividing the averaged p-p value of $2f$ PT signal for ten measurements by the 1σ noise. Revision has been made in the revised **Manuscript.**

Comment 2.9

- Line 163 mentions 'superb' common mode rejection. What is 'superb'? Also fig 4 c raises some questions especially for integration times over 100seconds. (The roll off increases and the variability is pronounced – why?)

Response:

'superb' means powerful ability of noise cancellation, which has been demonstrated numerically to be over 2 orders of magnitude in the **Supplementary Note 2.** We also mentioned this result in Response to **Comment 2.5.**

According to reference [1], for white phase noise, the equivalent degrees of freedom of Allan-Werle variance would be smaller for a longer integration time resulting in narrower confidence interval; the error then becomes relatively larger and the oscillation more serious.

Comments have been included in the **Supplementary Note 5. *Experimental details-Test of lower detection limit and long-term stability***

Comment 2.10

- Also – detection signal amplitude depends on probe and pump power levels, not to mention retaining the quadrature point. How stable are these over longer periods? This

leads into figure 5 – which needs clear straightforward explanation of what these results actually mean and why.

Response:

Thank you for pointing out this. In fact, we simultaneously detected the $2f$ signal (Fig.5(a)) from the lock-in and the slow-varying (we call it DC) component from the photodetector(not shown in Fig.3). The DC component was denoised by using Daubechies wavelet (db5) at level 6 and the denoised DC signal (V_{DC}) is shown as the blue curve in **Response Fig. 1**, while the p-p value of the $2f$ signal is shown as the red square dot. The p-p value shows a similar trend as the DC signal, which would include the effect of intensity fluctuation of probe as well as the drift of the operating point. Then, $2f$ signal was DC-compensated by multiplying it with a compensation factor ($\overline{V_{DC}}/V_{DC}$, $\overline{V_{DC}}$ is the average value of V_{DC} over three hours), and the p-p value of the compensated $2f$ signal is shown in Fig.5(c).

Response Figure 1 The p-p value of the $2f$ signal and the denoised DC signal.

The above comments have been included in the **Supplementary Note 5. Experimental details-Test of lower detection limit and long-term stability**

Comment 2.11

One final comment – the story could be significantly improved by a short, but convincing, exploration of longer term stability issues. These are mentioned but pump and probe powers and, for the probe, wavelength, are important. Also – whilst there are many comments on ‘common mode rejection’ there is no real evidence. What if the temperature of the cell is changed by, say, 50C?

Response:

Please see the response to **Comment 2.5**.

Comment 2.12

These modifications could, for this reviewer, certainly make the submission more accessible to potential readers. Whilst many of these potential ambiguities could in

principle be solved by sheer reader determination, it is definitely better to try to avoid this requirement. The concept overall is certainly interesting and worthy of publication.

Response:

We greatly appreciate the effort of the reviewer for giving so many useful comments.

Reference

- [1] Howe D. A., Allan D. U. & Barnes J. A. Properties of signal sources and measurement methods. *Proc. 35th Ann. Freq. Control Symposium* 669-716(1981).